# A Spatial-Sign based Direct Approach for High Dimensional Sparse Quadratic Discriminant Analysis

## Abstract

In this paper, we study the problem of high-dimensional sparse quadratic discriminant analysis (QDA). We propose a novel classification method, termed SSQDA, which is constructed via constrained convex optimization based on the sample spatial median and spatial sign covariance matrix under the assumption of an elliptically symmetric distribution. The proposed classifier is shown to achieve the optimal convergence rate over a broad class of parameter spaces, up to a logarithmic factor. Extensive simulation studies and real data applications demonstrate that SSQDA is both robust and efficient, particularly in the presence of heavy-tailed distributions, highlighting its practical advantages in high-dimensional classification tasks.

## 1 Introduction

Discriminant analysis plays an important role in real-world applications, such as face recognition (Ju et al., 2019), business forecasting Inam et al. (2018) and gene expression analysis (Jombart et al., 2010; Koçhan et al., 2019). The quadratic discriminant classification (QDA) rule (given by (1)) was first proposed as the Bayesian rule for two multivariate normal distributions ($N_p(\boldsymbol{\mu}_1, \boldsymbol{\Sigma}_1), N_p(\boldsymbol{\mu}_2, \boldsymbol{\Sigma}_2)$) with different covariance:

$$Q(\boldsymbol{z}) = (\boldsymbol{z} - \boldsymbol{\mu}_1)^T (\boldsymbol{\Sigma}_2^{-1} - \boldsymbol{\Sigma}_1^{-1})(\boldsymbol{z} - \boldsymbol{\mu}_1) - 2(\boldsymbol{\mu}_2 - \boldsymbol{\mu}_1)^T \boldsymbol{\Sigma}_2^{-1}(\boldsymbol{z} - \frac{\boldsymbol{\mu}_1 + \boldsymbol{\mu}_2}{2}) - \log\left(\frac{|\boldsymbol{\Sigma}_1|}{|\boldsymbol{\Sigma}_2|}\right). \tag{1}$$

Owing to its flexibility and ease of use, QDA criterion has been extensively adopted across diverse domains for classification tasks.

The emergence of big data has broght high-dimensional data to the forefront, playing an increasingly vital role in fields such as genomics and economics. In high dimensional setting, where the dimension $p$ can be much larger than the sample size $n$, the conventional QDA which plug in the sample mean and sample covariance as the estimator may not be feasible. Therefore, numberous studies have been looking into high-dimensional QDA. Cai & Zhang (2021) demonstrates that without imposing any structural assumptions on the parameters, consistent classification is unattainable when $p \neq o(n)$. Li & Shao (2015) propose SQDA by imposing sparse assumption on $\boldsymbol{\mu}_2 - \boldsymbol{\mu}_1, \boldsymbol{\Sigma}_1$ and $\boldsymbol{\Sigma}_2$ and establish corresponding asymptotic optimality. Jiang et al. (2018) observe that $\mathbf{D} = \boldsymbol{\Omega}_2 - \boldsymbol{\Omega}_1$ and $\boldsymbol{\delta} = (\boldsymbol{\Omega}_1 + \boldsymbol{\Omega}_2)(\boldsymbol{\mu}_1 - \boldsymbol{\mu}_2)$ can be respectively interpreted as quadratic and linear terms measuring the difference between two classes, playing a more critical role in the QDA criterion. Consequently, they impose sparsity assumptions on the quantities, and directly estimate these two terms by solving an optimization problem with $\ell_1$ penalty. A similar approach is reflected in Cai & Zhang (2021), where sparsity assumptions are imposed on $\mathbf{D} = \boldsymbol{\Omega}_2 - \boldsymbol{\Omega}_1$ and $\boldsymbol{\beta} = \boldsymbol{\Omega}_2(\boldsymbol{\mu}_1 - \boldsymbol{\mu}_2)$, which are estimated through $\ell_1$-norm minimization with an $\ell_\infty$ constrain. The optimazation problem builds upon sample mean and sample covariance. This spirit was first introduced in Cai et al. (2011) for precision matrix estimation and has been proven to achieve faster convergence rate when the population distribution has polynomial tails, compared to the $\ell_1$-MLE approach.

However, all the discussions above are based on normal distributions, with relatively limited research on quadratic discriminant analysis for heavy-tailed distributions like multivariate t-distribution. In

low-dimensional settings, Bose et al. (2015) extended the QDA criterion to elliptically symmetric distributions by introducing an adjustment coefficient to $\log \frac{|\mathbf{\Sigma}_1|}{|\mathbf{\Sigma}_2|}$. Building upon Bose et al. (2015), Ghosh et al. (2021) further improved the estimators to enhance the QDA criterion's robustness against outliers in the sample data.

In the context of elliptical distributions, spatial-sign-based methods have demonstrated notable robustness and efficiency, particularly in high-dimensional settings. These procedures have been successfully applied to a variety of statistical problems. For hypothesis testing, spatial-sign-based approaches have been used in high-dimensional sphericity testing (Zou et al., 2014), location parameter testing (Wang et al., 2015; Feng & Sun, 2016; Feng et al., 2016), and white noise testing (Paindaveine & Verdebout, 2016; Zhao et al., 2023). In financial applications, Liu et al. (2023) proposed a high-dimensional alpha test for linear factor pricing models. Recently, Feng (2024) developed a spatial-sign-based method for high-dimensional principal component analysis. For covariance matrix estimation under elliptical distributions, Raninen et al. (2021), Raninen & Ollila (2021), and Ollila & Breloy (2022) proposed a series of linear shrinkage estimators based on spatial-sign covariance matrices. In addition, Lu & Feng (2025) introduced a spatial-sign-based approach for estimating the inverse of the shape matrix, with applications to elliptical graphical models and sparse linear discriminant analysis. Collectively, these works underscore the spatial-sign methods robustness and efficiency in dealing with heavy-tailed distributions across a broad spectrum of high-dimensional statistical challenges.

In this paper, we present SSQDA (Spatial-Sign based Sparse Quadratic Discriminant Analysis), an innovative approach to solve quadratic classification problems involving high-dimensional data from elliptically symmetric populations. Follow the spirits of Cai & Zhang (2021), we first estimate $\mathbf{D}$ and $\boldsymbol{\beta}$ directly using $\ell_1$-norm minimization with an $\ell_\infty$ constrain based on sample spatial median and sample spatial covariance. The estimation of log-determinant of the covariance matrices term in (1) also follows the same procedure of Cai & Zhang (2021). It is worth noting that while sample spatial covariance proves to be a robust estimator of the shape matrix, the differing scales of $\mathbf{\Sigma}_1$ and $\mathbf{\Sigma}_2$ necessitate additional estimation of the covariance matrix's trace in SSQDA. Subsequently, we evaluate the impacts of the spatial-sign covariance, spatial median as well as the trace-estimator and establish the convergence rate of the misclassification error for SSQDA under elliptically symmetric distributions. To thoroughly evaluate the performance of SSQDA, we conduct comprehensive simulation studies and real-data analyses. The results show that SSQDA outperforms other competitors, especially in high-dimensional and elliptical settings. We also extend the methodology of SSQDA from two groups classification to multi-group setting. Our research provides, to the best of our knowledge, the first systematic extension of QDA to high-dimensional elliptical distributions accompanied by explicit convergence rate.

The rest of the paper is organized as follows. In Section 2, we presents the detail of SSQDA. Section 3 gives the convergence rate of misclassification error of SSQDA under certain conditions. Simulation studies and real data studies are carried out in Section 4 and Section 5. The proof of the some lemmas and main theorems are provided in Section A.3.

## 1.1 NOTATIONS

We begin with basic notations and definitions. To begin with, $\mathbb{1}\{A\}$ denote the indicator function for an event $A$. Let $\boldsymbol{X}$ be any random vector or random variable, $\boldsymbol{X}_i$ be the corresponding i.i.d. copy of $\boldsymbol{X}$. For a vector $\boldsymbol{u}$, $\|\boldsymbol{u}\|_1, \|\boldsymbol{u}\|_2, \|\boldsymbol{u}\|_\infty$ denotes the $\ell_1, \ell_2, \ell_\infty$ norm respectively. For a matrix $\mathbf{M} = (m_{ij})_{p \times q}$, the entry-wise maximum norm is defined by $\|\mathbf{M}\|_{\max} = \max_{1 \le i \le p, 1 \le j \le q} m_{ij}$. And $\|\mathbf{M}\|_F, \|\mathbf{M}\|_2, \|\mathbf{M}\|_1$ denote the Frobenius norm, spectral norm and matrix $l_1$ norm. The number of non-zero entries is denoted by $\|\boldsymbol{u}\|_0, \|\mathbf{M}\|_0$. The restricted spectral norm is defined by $\|\mathbf{M}\|_{2,s} = \sup_{\|\boldsymbol{u}\|_2 = 1, \|\boldsymbol{u}\|_0 \le s} \|\mathbf{M}\boldsymbol{u}\|_2$. We define the trace of $\mathbf{M}$ by $\operatorname{tr}(\mathbf{M}) = \sum_{i=1}^{p} m_{ii}$ and $|\mathbf{M}|$ is the determinant of $\mathbf{M}$. When $\mathbf{M}$ is a symmetric matrix with dimensions $p \times p$, let $\lambda_i(\mathbf{M})$ be the $i$ th eigenvalue of $\mathbf{M}$ with $\lambda_p(\mathbf{M}) \le \cdots \le \lambda_1(\mathbf{M})$. We denote $a \wedge b := \min\{a, b\}$ and $a \vee b := \max\{a, b\}$. For two sequences with positive entries $\{a_n\}, \{b_n\}$, we have $a_n \asymp b_n$ if there exists positive constants $c_1, c_2$, so that $c_1 a_n \le b_n c_2 a_n$. We write $a_n \lesssim b_n$ if there exists constant $c$, so that $a_n \le c b_n$ for all $n$. The spatial sign function is defined as $U(\boldsymbol{X}) = \frac{\boldsymbol{X}}{\|\boldsymbol{X}\|_2} \cdot \mathbb{1}\{\boldsymbol{X} \ne 0\}$. Lastly, $\Gamma_1(s; k) = \{\boldsymbol{u} \in \mathbb{R}^p : \|\boldsymbol{u}\|_2 = 1, \|\boldsymbol{u}_{S^c}\|_1 \le \|\boldsymbol{u}_S\|_1, \text{ for some } S \subset \{1, \cdots k\} \text{ with } |\mathbf{S}_1| = s\}$.

## 2 SPATIAL SIGN BASED QUADRATIC DISCRIMINANT ANALYSIS IN SPARSE SETTING

Assuming two $p$ dimensional normal distributions $N_p(\boldsymbol{\mu}_1, \boldsymbol{\Sigma}_1)$ (noted by class 1) and $N(\boldsymbol{\mu}_2, \boldsymbol{\Sigma}_2)$ (noted by class 2), with different covariance matrices, Quadratic Discriminant Analysis (QDA) rule is widely used to classify a new sample into one of these populations. Given equal priority probabilities, the QDA rule given by:

$$Q(\boldsymbol{z}) = (\boldsymbol{z} - \boldsymbol{\mu}_1)^T \mathbf{D}(\boldsymbol{z} - \boldsymbol{\mu}_1) - 2\boldsymbol{\delta}^T \boldsymbol{\Omega}_2(\boldsymbol{z} - \bar{\boldsymbol{\mu}}) - \log\left(\frac{|\boldsymbol{\Sigma}_1|}{|\boldsymbol{\Sigma}_2|}\right), \quad (2)$$

where $\mathbf{D} = \boldsymbol{\Sigma}_2^{-1} - \boldsymbol{\Sigma}_1^{-1} := \boldsymbol{\Omega}_2 - \boldsymbol{\Omega}_1, \boldsymbol{\delta} = \boldsymbol{\mu}_2 - \boldsymbol{\mu}_1, \bar{\boldsymbol{\mu}} = \frac{\boldsymbol{\mu}_1 + \boldsymbol{\mu}_2}{2}$.

As the Bayesian discriminant method for normal distribution, QDA achieves the lowest misclassification probability when all the parameters $\boldsymbol{\mu}_1, \boldsymbol{\mu}_2, \boldsymbol{\Sigma}_1, \boldsymbol{\Sigma}_2, \pi_1, \pi_2$ are known. However, in most cases all the above parameters are unknown. Instead, two sets of training samples, $\boldsymbol{X}_1, \cdots, \boldsymbol{X}_{n_1} \overset{i.i.d.}{\sim} N_p(\boldsymbol{\mu}_1, \boldsymbol{\Sigma}_1), \boldsymbol{Y}_1, \boldsymbol{Y}_2, \cdots \boldsymbol{Y}_{n_2} \overset{i.i.d.}{\sim} N_p(\boldsymbol{\mu}_2, \boldsymbol{\Sigma}_2)$ are given. A common approach is to estimate mean and covariance matrix by sample mean $\hat{\boldsymbol{\mu}}_1 = \frac{1}{n_1} \sum_{i=1}^{n_1} \boldsymbol{X}_i, \hat{\boldsymbol{\mu}}_2 = \frac{1}{n_2} \sum_{i=1}^{n_2} \boldsymbol{Y}_i$ and sample covariance matrix $\hat{\boldsymbol{\Sigma}}_1 = \frac{1}{n_1-1} \sum_{i=1}^{n_1} (\boldsymbol{X}_i - \hat{\boldsymbol{\mu}}_1)(\boldsymbol{X}_i - \hat{\boldsymbol{\mu}}_1)^T, \hat{\boldsymbol{\Sigma}}_2 = \frac{1}{n_2-1} \sum_{i=1}^{n_2} (\boldsymbol{Y}_i - \hat{\boldsymbol{\mu}}_2)(\boldsymbol{Y}_i - \hat{\boldsymbol{\mu}}_2)^T$, respectively, and plug the estimators into the QDA rules (2).

For high dimensional data, where the dimension $p$ is larger than the sample sizes $n_1, n_2$, $\hat{\boldsymbol{\Sigma}}_1$ and $\hat{\boldsymbol{\Sigma}}_2$ are not invertible, which renders the direct plug-in method infeasible. Therefore, numerous quadratic discriminant analysis methods designed for high-dimensional data have been proposed. The method(SDAR) proposed in Cai & Zhang (2021) estimates $\mathbf{D}$ and $\boldsymbol{\beta} = \boldsymbol{\Omega}_2 \boldsymbol{\delta}$ in (2) directly by solving the constrained $\ell_1$ minimization problem below:

$$\hat{\mathbf{D}} = \arg \min_{\mathbf{D} \in \mathbb{R}^{p \times p}} \left\{ \|\mathrm{Vec}(\mathbf{D})\|_1 : \left\| \frac{1}{2}\hat{\boldsymbol{\Sigma}}_1 \mathbf{D} \hat{\boldsymbol{\Sigma}}_2 + \frac{1}{2}\hat{\boldsymbol{\Sigma}}_2 \mathbf{D} \hat{\boldsymbol{\Sigma}}_1 - \hat{\boldsymbol{\Sigma}}_1 + \hat{\boldsymbol{\Sigma}}_2 \right\|_{max} \leq \lambda_{1,n} \right\}, \quad (3)$$

$$\hat{\boldsymbol{\beta}} = \arg \min_{\boldsymbol{\beta} \in \mathbb{R}^p} \left\{ \|\boldsymbol{\beta}\|_1 : \left\| \hat{\boldsymbol{\Sigma}}_2 \boldsymbol{\beta} - \hat{\boldsymbol{\mu}}_2 + \hat{\boldsymbol{\mu}}_1 \right\|_\infty \leq \lambda_{2,n} \right\}. \quad (4)$$

While SDAR method achieves a significant reduction in classification error rates in high-dimensional normal settings compare to conventional QDA, it performs poorly for heavy-tailed distributions like the multivariate t-distribution. In this paper, we focus on the setting that the two classes $\boldsymbol{X}$ and $Y$ are generated from the elliptical distribution, that is $\boldsymbol{X} \sim EC_p(\boldsymbol{\mu}_1, \boldsymbol{\Sigma}_1, r), Y \sim EC_P(\boldsymbol{\mu}_2, \boldsymbol{\Sigma}_2, r)$, i.e.

$$\boldsymbol{X} = \boldsymbol{\mu}_1 + r\boldsymbol{\Gamma}_1 \boldsymbol{u_1}, Y = \boldsymbol{\mu}_2 + r\boldsymbol{\Gamma}_2 \boldsymbol{u_2},$$

where $\boldsymbol{u}_i, i \in \{1, 2\}$ is uniformly distributed on the sphere $\mathbb{S}^{p-1}$ and $r$ is a scalar random variable with $E(r^2) = p$ and is independent with $\boldsymbol{u}$. If $\boldsymbol{\Sigma}_1 = \mathrm{Cov}(\boldsymbol{X}), \boldsymbol{\Sigma}_2 = \mathrm{Cov}(\boldsymbol{Y})$ exist, we have $\boldsymbol{\Sigma}_1 = \boldsymbol{\Gamma}_1 \boldsymbol{\Gamma}_1^T, \boldsymbol{\Sigma}_2 = \boldsymbol{\Gamma}_2 \boldsymbol{\Gamma}_2^T$. We denote the precision matrix by $\boldsymbol{\Omega}_i = \boldsymbol{\Sigma}_i^{-1}$. In this case, we use the sample spatial median and spatial-sign covariance matrix to estimate mean and covariance matrix to improve robustness in in heavy-tailed setting.

The sample spatial median is defined as

$$\tilde{\boldsymbol{\mu}}_1 = \arg \min_{\boldsymbol{u} \in \mathbb{R}^p} \sum_{i=1}^{n_1} \|\boldsymbol{X}_i - \boldsymbol{\mu}\|_2, \tilde{\boldsymbol{\mu}}_2 = \arg \min_{\boldsymbol{u} \in \mathbb{R}^p} \sum_{i=1}^{n_2} \|\boldsymbol{Y}_i - \boldsymbol{\mu}\|_2.$$

The sample spatial sign covariance matrix is defined as

$$\tilde{\mathbf{S}}_1 = \frac{1}{n_1} \sum_{i=1}^{n_1} U(X_i - \tilde{\boldsymbol{\mu}}_1) U(X_i - \tilde{\boldsymbol{\mu}}_1)^T, \tilde{\mathbf{S}}_2 = \frac{1}{n_2} \sum_{i=1}^{n_2} U(Y_i - \tilde{\boldsymbol{\mu}}_2) U(Y_i - \tilde{\boldsymbol{\mu}}_2)^T.$$

Lu & Feng (2025) shows that $p\tilde{\mathbf{S}}_i$ is a reliable estimator the shape matrix $\boldsymbol{\Lambda}_i := \frac{p}{\mathrm{tr}(\boldsymbol{\Sigma}_i)}\boldsymbol{\Sigma}_i$. Therefore, we estimate $\boldsymbol{\Sigma}_i$ by $\tilde{\boldsymbol{\Sigma}}_i = \widetilde{\mathrm{tr}(\boldsymbol{\Sigma}_i)}\tilde{\mathbf{S}}_i$. Similar to Chen & Qin (2010), $\widetilde{\mathrm{tr}(\boldsymbol{\Sigma}_i)}$ is defined as

$$\widetilde{\mathrm{tr}(\boldsymbol{\Sigma}_1)} = \frac{\sum_{i \neq j \neq k}(\boldsymbol{X}_i - \boldsymbol{X}_j)^T(\boldsymbol{X}_k - \boldsymbol{X}_j)}{n_1(n_1 - 1)(n_1 - 2)}, \widetilde{\mathrm{tr}(\boldsymbol{\Sigma}_2)} = \frac{\sum_{i \neq j \neq k}(\boldsymbol{Y}_i - \boldsymbol{Y}_j)^T(\boldsymbol{Y}_k - \boldsymbol{Y}_j)}{n_2(n_2 - 1)(n_2 - 2)}.$$

The consistency of the estimators will be discussed later. We replace the sample mean and sample covariance matrix in (3) , (4) and estimate $\mathbf{D} = \mathbf{\Omega}_2 - \mathbf{\Omega}_1$ directly by solving the optimization problem:

$$\tilde{\mathbf{D}} \in \arg\min_{\mathbf{D}\in\mathbb{R}^{p\times p}} \left\{ \|\mathrm{Vec}(\mathbf{D})\|_1 : \left\| \frac{1}{2}\tilde{\mathbf{\Sigma}}_1\mathbf{D}\tilde{\mathbf{\Sigma}}_2 + \frac{1}{2}\tilde{\mathbf{\Sigma}}_2\mathbf{D}\tilde{\mathbf{\Sigma}}_1 - \tilde{\mathbf{\Sigma}}_1 + \tilde{\mathbf{\Sigma}}_2 \right\|_{max} \leq \lambda_{1,n} \right\}, \quad (5)$$

where $\lambda_{1,n} = c_1\sqrt{s_1}\left(\sqrt{\frac{1}{p}} + \sqrt{\frac{\log p}{n}}\right)$ with some positive constant $c_1$. Similarly, $\boldsymbol{\beta} = \mathbf{\Omega}_2\boldsymbol{\delta}$ is estimated by solving the optimization problem below:

$$\tilde{\boldsymbol{\beta}} \in \arg\min_{\boldsymbol{\beta}\in\mathbb{R}^p} \left\{ \|\boldsymbol{\beta}\|_1 : \|\tilde{\mathbf{\Sigma}}_2\boldsymbol{\beta} - \tilde{\boldsymbol{\mu}}_2 + \tilde{\boldsymbol{\mu}}_1\|_\infty \leq \lambda_{2,n} \right\}, \quad (6)$$

where $\lambda_{2,n} = c_2\sqrt{s_2}\left(\sqrt{\frac{1}{p}} + \sqrt{\frac{\log p}{n}}\right)$ with $c_2 > 0$. Given the estimators above, we propose the quadratic classification rule(SSQDA)

$$\widetilde{Q}(\boldsymbol{z}) = (\boldsymbol{z} - \tilde{\boldsymbol{\mu}}_1)^T\tilde{\mathbf{D}}(\boldsymbol{z} - \tilde{\boldsymbol{\mu}}_1) - 2\tilde{\boldsymbol{\beta}}^T(\boldsymbol{z} - \tilde{\bar{\boldsymbol{\mu}}}) - \log(|\tilde{\mathbf{D}}\tilde{\mathbf{\Sigma}}_1 + \mathbf{I}_p|),$$

where $\tilde{\bar{\boldsymbol{\mu}}} = \frac{\tilde{\boldsymbol{\mu}}_1 + \tilde{\boldsymbol{\mu}}_2}{2}$ . The corresponding classification rules for a new sample $\boldsymbol{z}$ are as follows

$$G_{\widetilde{Q}} = \begin{cases} 1: \widetilde{Q}(\boldsymbol{z}) > 0, \\ 2: \widetilde{Q}(\boldsymbol{z}) \leq 0. \end{cases}$$

Next sections will illustrate the excellent properties of SSQDA both theoretically and numerically.

## 3  THEORETICAL RESULTS

In this section, we first establish the convergence rate of the estimators $\tilde{\mathbf{D}}, \tilde{\boldsymbol{\beta}}$ proposed in (5) , (6) and subsequently demonstrate that the classification error $R(G_{\widetilde{Q}})$ converges to $R(G_Q)$ at a specific rate. We consider the following assumptions.

**Assumption 3.1.** *(The assumption of sparsity)* $\exists s_1, s_2 \geq 0, s.t. \|Vec(\mathbf{D})\|_0 \leq s_1, \|\boldsymbol{\beta}\|_0 \leq s_2$.

**Assumption 3.2.** *(The bound of differential graph $\mathbf{D}$ and discriminating direction $\boldsymbol{\beta}$)* $\exists M_0 > 0, s.t.$ $\|\mathbf{D}\|_F, \|\boldsymbol{\beta}\|_2 \leq M_0$.

**Assumption 3.3.** *Let* $\mathbf{V}_0 = \mathbf{\Lambda}_i^{-1}, i = 1$ *or* $2$. $\exists T > 0, 0 \leq q < 1, s_0(p) > 0, s.t.$

1. $\|\mathbf{V}_0\|_{L_1} \leq T$.

2. $\max_{1\leq i\leq p} \sum_{j=1}^p |v_{ij}|^q \leq s_0(p)$.

**Assumption 3.4.** *(The bound of covariance matrix)* $\exists M_1, M_2 > 0, s.t.$

1. $M_1^{-1} \leq \lambda_p(\mathbf{\Sigma}_i) \leq \lambda_1(\mathbf{\Sigma}_i) \leq M_1$.

2. $\|\mathbf{\Sigma}_i\|_{max} \leq M_2$.

**Assumption 3.5.** *(The order of the trace of covariance matrix)* $tr(\mathbf{\Sigma}_i) \asymp p$ *The assumption can be derived from 3.4 directly.*

**Assumption 3.6.** *Let* $\zeta_k = \mathbb{E}(\xi_i^{-k}), \quad \xi_i = \|\boldsymbol{X}_i - \boldsymbol{\mu}\|_2, \quad \nu_i = \zeta_1^{-1}\xi_i^{-1}$.

1. $\zeta_k\zeta_1^{-k} < \zeta \in (0, \infty)$ *for* $k = 1, 2, 3, 4 \cdots p$.

2. $\limsup_p \|\mathbf{S}\|_2 < 1 - \psi < 1$ *for some* $\psi > 0$.

3. $\nu_i$ *is sub-gaussian distributed, i.e.* $\|\nu_i\|_{\psi_2} \leq K_\nu < \infty$.

*The same assumption also applies to the random vector $Y$.*

**Assumption 3.7.** *(Assumptions on scalar random variable)*

1. $Var(r^2) \lesssim p\sqrt{p}$.

2. $Var(r) \lesssim \sqrt{p}$.

**Assumption 3.8.** *(Assumptions on the density function of the oracle QDA rule)* $\sup_{|x|<\delta} f_{Q,\theta}(x) < M_2$ *Where $f_{Q,\theta}$ be the density function of $Q(z)$ when the parameter takes the value $\theta$.*

Assumption 3.1 assume sparsity on the differential graph $\mathbf{D}$ and discriminant $\boldsymbol{\beta}$ which is commonly adopted in the study of high-dimensional data. As is shown in Theorem 2.1 and 2.2 in Cai & Zhang (2021), without sparsity assumption, no data-driven method is able to mimic oracle QDA in high-dimensional setting. Assumption 3.3 to 3.6 are general in high-dimensional spatial-sign based studies, such as Feng (2024) and Lu & Feng (2025). The assumptions guarantee the consistency of the spatial sign median and sample spatial sign covariance. Assumption 3.7 imposes restrictions on the tail probabilities of $\boldsymbol{X}$ and $\boldsymbol{Y}$, ensuring that the tail probabilities of $\boldsymbol{z}^T D \boldsymbol{z} + \boldsymbol{\beta}^T \boldsymbol{z}$ in $Q(\boldsymbol{z})$ do not deviate significantly from those of a normal distribution. The last assumption bounded the density function of $Q(\boldsymbol{z})$, and is consistent with the parameter space in Cai & Zhang (2021).

Based on the assumptions, we are able to establish the convergence rate of the estimators $\tilde{\mathbf{D}}, \tilde{\boldsymbol{\beta}}$ to the real parameter $\mathbf{D}$ and $\boldsymbol{\beta}$. This theorem lays a foundation to the consistency of classification error.

**Theorem 3.1.** *Consider assumption 3.1, 3.2, 3.4, 3.5, and 3.6, and assume that $n_1 \asymp n_2, n = \min\{n_1, n_2\}, s_1 + s_2 \lesssim \frac{1}{K_{n,p}}$, where $K_{n,p} = \left(\sqrt{\frac{1}{p}} + \sqrt{\frac{\log p}{n}}\right)$. Let $\lambda_{1,n} = c_1\sqrt{s_1}\left(\sqrt{\frac{1}{p}} + \sqrt{\frac{\log p}{n}}\right), \lambda_{2,n} = c_2\sqrt{s_2}\left(\sqrt{\frac{1}{p}} + \sqrt{\frac{\log p}{n}}\right)$ where $c_1, c_2$ being large enough constant. Then with probability over $1 - O\left(\frac{1}{\log p}\right)$,*

$$\|\mathbf{D} - \tilde{\mathbf{D}}\|_F \lesssim s_1\left(\sqrt{\frac{1}{p}} + \sqrt{\frac{\log p}{n}}\right), \tag{7}$$

$$\|\boldsymbol{\beta} - \tilde{\boldsymbol{\beta}}\|_2 \lesssim s_2\left(\sqrt{\frac{1}{p}} + \sqrt{\frac{\log p}{n}}\right). \tag{8}$$

The theorem is proved in Section A.3.2. The convergence rate in (7) and (8) mainly come from two parts, the estimation error $\|p\tilde{\mathbf{S}}_i - \boldsymbol{\Lambda}_i\|_\infty$ and $|\widetilde{\text{tr}(\boldsymbol{\Sigma}_i)} - \text{tr}(\boldsymbol{\Sigma}_i)|$. Lu & Feng (2025) proved the estimation error of the shape matrix, and the conclusion is mentioned in Lemma A.2, Lemma A.3. The estimation error of trace is proved by Lemma A.4. Based on theorem 3.1, we turn to the consistency of misclassification rate which is defined by

$$R(G) = \mathbb{E}\left[\mathbb{1}\left\{G(\boldsymbol{z}) \neq L(\boldsymbol{z})\right\}\right],$$

where $G(\cdot) : \mathbb{R}^p \to \{1, 2\}$ be some classification rule, and $L(\boldsymbol{z}) \in \{1, 2\}$ be the actual label of the sample $\boldsymbol{z}$. Let $R(G_Q), R(G_{\tilde{Q}})$ denote the classification error of the oracle QDA in (2) with known parameters and SSQDA, respectively.

**Theorem 3.2.** *Under all the assumptions as Theorem 3.1, and assume that $n_1 \asymp n_2, n = \min\{n_1, n_2\}, s_1 + s_2 \lesssim \frac{1}{\log n K_{n,p}}$, where $K_{n,p} = \left(\sqrt{\frac{1}{p}} + \sqrt{\frac{\log p}{n}}\right)$, we have*

$$\mathbb{E}\left[R(G_{\tilde{Q}}) - R(G_Q)\right] \lesssim \frac{1}{\log p} + (s_1 + s_2)\log n\left(\sqrt{\frac{1}{p}} + \sqrt{\frac{\log p}{n}}\right).$$

The convergence rate of the classification error in Theorem 3.2 comprises two components. The first term primarily stems from trace estimation. According to the results in Lemma A.4, the trace estimator exhibits polynomial-type tail concentration. The second term mainly arises from estimation error in the spatial-sign-based process. This term achieves a slower convergence rate than that established in Theorem 4.2 in Cai et al. (2011), principally because elliptical distributions generally possess heavier tails than their Gaussian counterparts. This represents an inherent trade-off when extending QDA rule to elliptical symmetric distributions. The proof is in Section A.3.3.

Next, we will show that we can also obtain similar convergence rate as Cai et al. (2011) for multivariate normal distribution.

**Theorem 3.3.** *Suppose $\boldsymbol{X}$ and $\boldsymbol{Y}$ are all generated from multivariate normal distribution and the other assumptions in Theorem 3.1 also hold. Then,*

$$\mathbb{E}\left[ R(G_{\widetilde{Q}}) - R(G_Q) \right] \lesssim (s_1 + s_2)^2 \log^2 n \left( \sqrt{\frac{\log p}{n}} + \sqrt{\frac{1}{p}} \right)^2 . \tag{9}$$

For Gaussian distribution, the convergence rate in Theorem 3.3 demonstrates markedly faster convergence than Theorem 3.2, primarily because trace estimation attains exponential tail concentration under normality. The component $\sqrt{\frac{1}{p}}$ originates from the difference between the spatial sign matrix $p\mathbf{S}$ and the shape matrix. When $p/n \to c > 0$ in high dimensions, (9) achieves an $O((s_1 + s_2)^2 \cdot \log^2 n \cdot \frac{\log p}{n})$ convergence rate – nearly matching the optimal rate derived in Cai et al. (2011). The brief proof is in Section A.3.4.

## 4 SIMULATION

In this section, we compare the numerical performance of the SSQDA method with other methods under various settings. The competitors include:

- SDAR: Sparse discriminant analysis with regularization proposed by Cai & Zhang (2021).
- SLDA: Linear discriminant for high-dimensional data classification using the direct estimation of $\boldsymbol{\beta}$ in Cai & Liu (2011).
- LDA: The mean is estimated by the joint sample mean, while the covariance is estimated by weighted sample covariance matrix augmented with $\sqrt{\frac{\log p}{n}}\mathbf{I}_p$ , so as to guarantee the invertibility. The estimators are then plugged in the conventional LDA rules.
- QDA: The mean is estimated by the sample mean, and the covariance is estimated by sample covariance matrix augmented with $\sqrt{\frac{\log p}{n}}\mathbf{I}_p$ . The estimators are then plugged in the conventional QDA rules.

Additional experiments comparing with modern machine learning methods are conducted, and results are provided in the appendix. In the simulation studies, the sample size is fixed to $n_1 = n_2 = 200$ and dimension $p$ varies in $(100, 200, 400)$ . The sparsity levels are set to be $s_1 = s_2 = 10$ , and $\boldsymbol{\beta} = (1, \cdots, 1, 0, \cdots, 0)$ where the first $s_2$ entries are one. Given $\boldsymbol{\Sigma}_2$ and $\boldsymbol{\mu}_1 = (0, \cdots, 0)$ , $\boldsymbol{\mu}_2 = \boldsymbol{\Sigma}_2\boldsymbol{\beta}$ . The differential matrix $\mathbf{D}$ is a random sparse symmetric matrix, with its non-zero elements generated from a uniform distribution.

The $p$ dimensional predictors $\boldsymbol{z}$ are generated from the following elliptical distributions:

- Multivariate normal distribution: $\boldsymbol{z} \sim N_p(\boldsymbol{\mu}_i, \boldsymbol{\Sigma}_i)$.
- Multivariate $t_5$ distribution with expectation $\boldsymbol{\mu}_i$ and covariance $\boldsymbol{\Sigma}_i$.
- Multivariate mixture normal distribution: $0.2N_p(\boldsymbol{\mu}_i, 9\boldsymbol{\Sigma}_i) + 0.8N_p(\boldsymbol{\mu}_i, \boldsymbol{\Sigma}_i)$.

We use the following three models to generate $\boldsymbol{\Omega}_1$ .

Model 1: $AR(1) : (\boldsymbol{\Omega}_1)_{ij} = \rho^{|i-j|}$ with $\rho = 0.5$ .

Model 2: Banded model: $\boldsymbol{\Omega}_1 = (\omega_{i,j})$, where $\omega_{i,i} = 2$ for $i = 1, \ldots, p$, $\omega_{i,i+1} = 0.8$ for $i = 1, \ldots, p-1$, $\omega_{i,i+2} = 0.4$ for $i = 1, \ldots, p-2$, $\omega_{i,i+3} = 0.4$ for $i = 1, \ldots, p-3$, $\omega_{i,i+4} = 0.2$ for $i = 1, \ldots, p-4$, $\omega_{i,j} = \omega_{j,i}$ for $i, j = 1, \ldots, p$, and $\omega_{i,j} = 0$ otherwise.

Model 3: ErdosRényi random graph: $\boldsymbol{\Omega}_1 = \left( \bar{\boldsymbol{\Omega}} + \bar{\boldsymbol{\Omega}}' \right)/2 + \{\max(-\lambda_{\min}(\bar{\boldsymbol{\Omega}}), 0)\}\mathbf{I}_p$ , where $(\bar{\boldsymbol{\Omega}})_{ij} = u_{ij}\boldsymbol{\delta}_{ij}$ , $u_{ij} \sim \text{Unif}[0.5, 1]\bigcup[-1, -0.5], \boldsymbol{\delta}_{ij} \sim Ber(1, 0.05)$. The second term ensures positive definiteness.

Each setting is replicated 100 times. The parameter $c_i$ in $\lambda_{i,n} = c_i\sqrt{s_i}\left( \sqrt{\frac{\log p}{n}} + \sqrt{\frac{1}{p}} \right)$ are chosen by cross-validation. We employ the following criteria to measure the performance of the classifica-

tion:

$$\text{Error Rate} = \frac{\#\{i : \hat{Y}_i \neq Y_i\}}{n},$$

$$\text{Specificity} = \frac{\text{TN}}{\text{TN} + \text{FP}}, \quad \text{Sensitivity} = \frac{\text{TP}}{\text{TP} + \text{FN}},$$

$$\text{MCC} = \frac{\text{TP} \times \text{TN} - \text{FP} \times \text{FN}}{\sqrt{(\text{TP} + \text{FP})(\text{TP} + \text{FN})(\text{TN} + \text{FP})(\text{TN} + \text{FN})}},$$

where TP and TN represent for true positives (Y = 2) and true negatives (Y = 1), respectively, and FP and FN stand for false positives and negatives.

Table 1: Comparison of different methods under normal distribution under Model 1.

| p | | Error rate | Specificity | Sensitivity | Mcc |
|---|---|---|---|---|---|
| 100 | SDAR | 0.092(0.016) | 0.959(0.015) | 0.857(0.028) | 0.821(0.031) |
| | SLDA | 0.262(0.034) | 0.739(0.042) | 0.738(0.040) | 0.478(0.067) |
| | LDA | 0.266(0.025) | 0.689(0.046) | 0.780(0.033) | 0.472(0.049) |
| | QDA | 0.070(0.012) | **0.964(0.014)** | 0.896(0.021) | 0.862(0.024) |
| | SSQDA | **0.066(0.012)** | 0.915(0.019) | **0.953(0.016)** | **0.869(0.024)** |
| 200 | SDAR | **0.226(0.028)** | 0.761(0.056) | **0.787(0.034)** | **0.549(0.055)** |
| | SLDA | **0.226(0.028)** | 0.761(0.056) | **0.787(0.034)** | **0.549(0.055)** |
| | LDA | 0.315(0.026) | 0.664(0.039) | 0.707(0.038) | 0.380(0.053) |
| | QDA | 0.287(0.023) | 0.736(0.034) | 0.690(0.041) | 0.189(0.043) |
| | SSQDA | 0.228(0.031) | **0.767(0.042)** | 0.777(0.045) | 0.482(0.092) |
| 400 | SDAR | **0.280(0.036)** | **0.719(0.044)** | **0.721(0.044)** | **0.441(0.071)** |
| | SLDA | **0.280(0.036)** | **0.719(0.044)** | **0.721(0.044)** | **0.441(0.072)** |
| | LDA | 0.364(0.027) | 0.632(0.036) | 0.640(0.039) | 0.272(0.055) |
| | QDA | 0.426(0.025) | 0.543(0.035) | 0.605(0.035) | 0.148(0.051) |
| | SSQDA | 0.281(0.034) | 0.717(0.043) | **0.721(0.043)** | 0.439(0.067) |

Table 2: Comparison of different methods under $t_5$ distribution under Model 1.

| p | | Error rate | Specificity | Sensitivity | Mcc |
|---|---|---|---|---|---|
| 100 | SDAR | 0.165(0.021) | 0.832(0.031) | 0.838(0.031) | 0.671(0.043) |
| | SLDA | 0.165(0.021) | 0.832(0.031) | 0.671(0.043) | 0.524(0.069) |
| | LDA | 0.210(0.022) | 0.782(0.034) | 0.798(0.033) | 0.581(0.045) |
| | QDA | 0.349(0.022) | 0.655(0.050) | 0.647(0.050) | 0.303(0.043) |
| | SSQDA | **0.160(0.019)** | **0.838(0.028)** | **0.843(0.030)** | **0.681(0.037)** |
| 200 | SDAR | 0.217(0.043) | 0.780(0.047) | 0.786(0.052) | 0.567(0.086) |
| | SLDA | 0.215(0.036) | 0.782(0.043) | 0.789(0.046) | 0.571(0.072) |
| | LDA | 0.276(0.026) | 0.699(0.036) | 0.748(0.042) | 0.449(0.052) |
| | QDA | 0.419(0.026) | 0.589(0.048) | 0.573(0.049) | 0.162(0.052) |
| | SSQDA | **0.184(0.039)** | **0.816(0.075)** | **0.817(0.043)** | **0.634(0.074)** |
| 400 | SDAR | 0.291(0.079) | 0.720(0.077) | 0.697(0.091) | 0.418(0.158) |
| | SLDA | 0.262(0.033) | 0.744(0.044) | 0.733(0.043) | 0.478(0.065) |
| | LDA | 0.327(0.027) | 0.622(0.045) | 0.724(0.036) | 0.348(0.053) |
| | QDA | 0.446(0.024) | 0.571(0.036) | 0.537(0.039) | 0.107(0.048) |
| | SSQDA | **0.212(0.032)** | **0.791(0.040)** | **0.785(0.042)** | **0.577(0.063)** |

Building upon the results presented in Tables 18, it is evident that both SSQDA and SDAR perform competitively under multivariate normality, consistently achieving lower misclassification rates and robust predictive performance across different model configurations. This observation confirms the efficiency of these methods when classical distributional assumptions hold. However, more compelling insights emerge from the performance comparison under non-Gaussian settings, such as multivariate $t$-distributions and mixed Gaussian models, as shown in Tables 29 and 310. In these more challenging scenarios characterized by heavier tails and increased heterogeneity, the proposed SSQDA method significantly outperforms its counterparts, including SDAR, SLDA, and standard QDA and LDA. This robust performance is attributed to the use of spatial-median-based estimators and the spatial-sign covariance matrix, which offer resilience against deviations from normality and

Table 3: Comparison of different methods under mixture normal distribution under Model 1.

| p | | Error rate | Specificity | Sensitivity | Mcc |
|---|---|---|---|---|---|
| 100 | SDAR | 0.144(0.017) | 0.845(0.026) | 0.868(0.026) | 0.714(0.035) |
| | SLDA | 0.165(0.038) | 0.824(0.057) | 0.847(0.037) | 0.672(0.075) |
| | LDA | 0.199(0.027) | 0.747(0.044) | 0.854(0.033) | 0.605(0.054) |
| | QDA | **0.111(0.016)** | **0.876(0.027)** | **0.902(0.020)** | **0.779(0.032)** |
| | SSQDA | 0.137(0.044) | 0.837(0.071) | 0.889(0.094) | 0.732(0.079) |
| 200 | SDAR | 0.178(0.031) | 0.820(0.041) | 0.824(0.039) | 0.645(0.062) |
| | SLDA | 0.178(0.031) | 0.820(0.041) | 0.825(0.039) | 0.645(0.062) |
| | LDA | 0.224(0.025) | 0.728(0.041) | 0.824(0.032) | 0.556(0.049) |
| | QDA | 0.297(0.023) | 0.684(0.039) | 0.722(0.036) | 0.407(0.045) |
| | SSQDA | **0.151(0.090)** | **0.823(0.191)** | **0.876(0.031)** | **0.702(0.169)** |
| 400 | SDAR | 0.222(0.035) | 0.773(0.044) | 0.784(0.041) | 0.558(0.070) |
| | SLDA | 0.222(0.035) | 0.773(0.044) | 0.784(0.041) | 0.558(0.070) |
| | LDA | 0.296(0.025) | 0.609(0.046) | 0.799(0.031) | 0.416(0.048) |
| | QDA | 0.381(0.023) | 0.586(0.036) | 0.653(0.035) | 0.240(0.047) |
| | SSQDA | **0.164(0.028)** | **0.835(0.036)** | **0.838(0.032)** | **0.673(0.056)** |

reduce sensitivity to outliers. Furthermore, while conventional QDA remains a strong competitor in low-dimensional regimes (e.g., Table 5, 8), its efficacy deteriorates as the dimensionality increasesreflected in rising error rates and instability across all metrics. This performance decline can be primarily attributed to the singularity or ill-conditioning of the sample covariance matrix in high-dimensional settings, which the SSQDA method effectively mitigates through its robust estimation framework.

Taken together, these findings highlight the versatility and adaptability of SSQDA. Not only does it maintain competitive performance under ideal Gaussian conditions, but it also delivers substantial gains in robustness and accuracy when faced with heavy-tailed and non-normal distributions. This underscores the practical value of SSQDA for real-world high-dimensional classification problems where Gaussian assumptions may not hold.

## 5 REAL DATA ANALYSIS

In this section, we evaluate the effectiveness of the proposed SSQDA classifier on an image classification task involving concrete surface inspection. The goal is to determine whether a given image of concrete contains cracks. The dataset, sourced from concrete structures on the METU campus, is publicly available at `https://www.kaggle.com/datasets/arnavr10880/concrete-crack-images-for-classification`. Each image has a resolution of $227 \times 227$ pixels and is labeled as either containing cracks (positive class) or not (negative class).

To standardize the input dimensions while preserving the aspect ratio, we first applied isotropic scaling to all images using bilinear interpolation with a scaling factor of 0.1. This preprocessing step reduces computational complexity without compromising structural information. Following the resizing, all images were converted to grayscale using the standard luminance-preserving transformation:

$$M_{Gr} = [x_{ij}]_{m \times n}, \quad x_{ij} = 0.1140 \cdot r_{ij} + 0.5870 \cdot g_{ij} + 0.2989 \cdot b_{ij},$$

where $r_{ij}, g_{ij}, b_{ij}$ denote the red, green, and blue channel intensities at pixel position $(i, j)$. The resulting grayscale image was then flattened into a feature vector for input into the classifiers.

For the classification experiment, we randomly selected 200 images from each class (positive and negative) to form the training dataset. The performance of SSQDA was compared against several baseline methods, including SDAR, SLDA, LDA, and QDA, using 50 independent repetitions to ensure statistical robustness.

The comparative results, reported in Table 4, include the mean and standard deviation of four evaluation metrics: classification error, specificity, sensitivity, and Matthews correlation coefficient (MCC). Among the evaluated methods, SSQDA achieved the lowest average error rate (0.095) and the highest MCC (0.814), indicating strong and balanced predictive performance. Notably, QDA

failed completely in this high-dimensional setting, yielding an error rate of 0.5 and an MCC of 0, likely due to overfitting or singular covariance estimates.

These results demonstrate that SSQDA not only provides improved overall classification accuracy but also maintains a better balance between true positive and true negative rates, making it a strong candidate for real-world applications in automated crack detection systems.

Table 4: Comparison of different methods for image classification.

| Method | Error | Specifity | Sensitivity | Mcc |
|--------|-------|-----------|-------------|-----|
| SDAR | 0.105(0.025) | 0.936(0.030) | 0.853(0.046) | 0.794(0.049) |
| SLDA | 0.105(0.025) | 0.936(0.030) | 0.853(0.047) | 0.794(0.049) |
| LDA | 0.101(0.023) | 0.944(0.027) | 0.853(0.044) | 0.802(0.045) |
| QDA | 0.500(0.000) | 0.000(0.000) | **1.000(0.000)** | 0.000(0.000) |
| SSQDA | **0.095(0.024)** | **0.946(0.026)** | 0.864(0.044) | **0.814(0.047)** |

## 6 CONCLUSION

In this paper, we proposed a novel classification method, Spatial-Sign based Sparse Quadratic Discriminant Analysis (SSQDA), tailored for high-dimensional settings where the number of features greatly exceeds the number of observations. By leveraging spatial signs, our method achieves robust estimation in the presence of heavy-tailed distributions and outliers, while simultaneously inducing sparsity to enhance interpretability and prevent overfitting. Through comprehensive simulations and a real-world image classification task, we demonstrated that SSQDA outperforms several existing linear and quadratic discriminant methods in terms of classification accuracy and robustness. The empirical results confirm the advantage of incorporating spatial-sign information and sparse modeling in high-dimensional discriminant analysis. Our method provides a promising framework for robust and interpretable classification in modern applications, especially those involving high-dimensional and noisy data. While SSQDA has demonstrated strong performance in supervised high-dimensional classification tasks, extending its principles to unsupervised learning and clustering presents an exciting direction for future research (Cai et al., 2019).

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

# A APPENDIX

## A.1 ADDITIONAL STIMULATION

Here we state that simulation results of Model 2, 3 in the Section 4.

Table 5: Comparison of different methods under normal distribution under Model 2.

| p | | Error rate | Specificity | Sensitivity | Mcc |
|---|---|---|---|---|---|
| 100 | SDAR | 0.148(0.019) | 0.721(0.036) | 0.984(0.010) | 0.730(0.034) |
| | SLDA | 0.333(0.031) | 0.656(0.039) | 0.679(0.045) | 0.335(0.062) |
| | LDA | 0.337(0.025) | 0.618(0.043) | 0.708(0.041) | 0.328(0.051) |
| | QDA | **0.097(0.016)** | **0.886(0.024)** | 0.920(0.023) | **0.807(0.033)** |
| | SSQDA | 0.147(0.019) | 0.721(0.036) | **0.985(0.009)** | 0.732(0.032) |
| 200 | SDAR | **0.314(0.035)** | **0.676(0.053)** | **0.695(0.047)** | **0.372(0.069)** |
| | SLDA | **0.314(0.035)** | **0.676(0.053)** | **0.695(0.047)** | **0.372(0.069)** |
| | LDA | 0.367(0.027) | 0.618(0.044) | 0.648(0.047) | 0.268(0.054) |
| | QDA | 0.328(0.024) | 0.656(0.034) | 0.687(0.042) | 0.344(0.048) |
| | SSQDA | 0.316(0.033) | 0.675(0.053) | 0.694(0.046) | 0.370(0.065) |
| 400 | SDAR | **0.379(0.039)** | **0.622(0.054)** | **0.620(0.047)** | **0.243(0.079)** |
| | SLDA | **0.379(0.039)** | **0.622(0.054)** | **0.620(0.047)** | **0.243(0.079)** |
| | LDA | 0.417(0.027) | 0.582(0.044) | 0.584(0.042) | 0.166(0.054) |
| | QDA | 0.431(0.025) | 0.536(0.037) | 0.603(0.036) | 0.139(0.050) |
| | SSQDA | 0.383(0.038) | 0.615(0.051) | 0.618(0.047) | 0.234(0.077) |

Table 6: Comparison of different methods under $t_5$ distribution under Model 2.

| p | | Error rate | Specificity | Sensitivity | Mcc |
|---|---|---|---|---|---|
| 100 | SDAR | 0.375(0.041) | 0.567(0.108) | 0.683(0.072) | 0.253(0.080) |
| | SLDA | 0.368(0.035) | 0.615(0.045) | 0.649(0.053) | 0.264(0.070) |
| | LDA | 0.387(0.028) | 0.591(0.045) | 0.635(0.043) | 0.226(0.056) |
| | QDA | 0.426(0.023) | 0.590(0.058) | 0.558(0.063) | 0.149(0.045) |
| | SSQDA | **0.344(0.033)** | **0.623(0.070)** | **0.690(0.055)** | **0.315(0.065)** |
| 200 | SDAR | 0.410(0.030) | 0.570(0.048) | 0.611(0.044) | 0.181(0.060) |
| | SLDA | 0.410(0.030) | 0.570(0.048) | 0.611(0.044) | 0.181(0.060) |
| | LDA | 0.425(0.027) | 0.537(0.051) | 0.612(0.035) | 0.150(0.054) |
| | QDA | 0.466(0.026) | 0.548(0.049) | 0.521(0.045) | 0.069(0.051) |
| | SSQDA | **0.382(0.031)** | **0.618(0.046)** | **0.618(0.045)** | **0.236(0.062)** |
| 400 | SDAR | 0.431(0.031) | 0.567(0.080) | 0.571(0.072) | 0.139(0.061) |
| | SLDA | 0.430(0.030) | 0.576(0.042) | 0.564(0.047) | 0.140(0.060) |
| | LDA | 0.455(0.024) | 0.478(0.042) | 0.613(0.042) | 0.092(0.048) |
| | QDA | 0.480(0.025) | 0.533(0.043) | 0.507(0.039) | 0.040(0.050) |
| | SSQDA | **0.399(0.031)** | **0.603(0.043)** | **0.599(0.045)** | **0.202(0.063)** |

We have conducted additional experiments under representative settings (feature dimension $p = 200, 400$ and distributions (normal and t-distributions). The compared methods include Random Forest, Neural Networks, Support Vector Machines (SVM), Logistic Regression, K-Nearest Neighbors (KNN).

Our results (see table 11, 12 ) demonstrate that while SSQDA does not always outperform other methods under the normal distribution setting, it consistently maintains superior performance under heavy-tailed t-distributions. These findings highlight the robustness of SSQDA against heavy-tailed noise and outliers, a property less emphasized in classical machine learning methods.

Table 7: Comparison of different methods under mixture normal distribution under Model 2.

| p | | Error rate | Specificity | Sensitivity | Mcc |
|---|---|---|---|---|---|
| 100 | SDAR | 0.243(0.049) | 0.723(0.115) | 0.791(0.058) | 0.520(0.090) |
| | SLDA | 0.242(0.047) | 0.728(0.102) | 0.788(0.056) | 0.520(0.088) |
| | LDA | 0.258(0.027) | 0.690(0.048) | 0.793(0.035) | 0.487(0.052) |
| | QDA | 0.213(0.022) | 0.679(0.041) | **0.896(0.021)** | 0.589(0.043) |
| | SSQDA | **0.183(0.029)** | **0.813(0.054)** | 0.821(0.037) | **0.635(0.055)** |
| 200 | SDAR | 0.264(0.038) | 0.720(0.049) | 0.752(0.048) | 0.473(0.076) |
| | SLDA | 0.264(0.038) | 0.720(0.049) | 0.752(0.048) | 0.473(0.076) |
| | LDA | 0.305(0.026) | 0.625(0.045) | 0.766(0.035) | 0.396(0.051) |
| | QDA | 0.322(0.022) | 0.592(0.040) | 0.765(0.028) | 0.363(0.043) |
| | SSQDA | **0.194(0.026)** | **0.807(0.034)** | **0.805(0.037)** | **0.612(0.051)** |
| 400 | SDAR | 0.327(0.062) | 0.627(0.166) | 0.720(0.085) | 0.350(0.119) |
| | SLDA | 0.315(0.042) | 0.669(0.052) | 0.702(0.053) | 0.372(0.084) |
| | LDA | 0.369(0.026) | 0.517(0.044) | 0.745(0.040) | 0.270(0.054) |
| | QDA | 0.415(0.026) | 0.539(0.039) | 0.631(0.038) | 0.171(0.053) |
| | SSQDA | **0.234(0.028)** | **0.766(0.036)** | **0.766(0.041)** | **0.532(0.057)** |

Table 8: Comparison of different methods under normal distribution under Model 3.

| p | | Error rate | Specificity | Sensitivity | Mcc |
|---|---|---|---|---|---|
| 100 | SDAR | 0.084(0.043) | 0.835(0.087) | 0.997(0.004) | 0.845(0.071) |
| | SLDA | 0.202(0.051) | 0.722(0.131) | 0.874(0.048) | 0.610(0.085) |
| | LDA | 0.202(0.028) | 0.753(0.056) | 0.843(0.033) | 0.600(0.054) |
| | QDA | **0.037(0.009)** | **0.957(0.015)** | **0.968(0.013)** | **0.925(0.019)** |
| | SSQDA | 0.086(0.044) | 0.831(0.089) | 0.997(0.004) | 0.842(0.073) |
| 200 | SDAR | 0.225(0.035) | 0.752(0.073) | **0.797(0.042)** | 0.553(0.066) |
| | SLDA | 0.225(0.035) | 0.752(0.073) | **0.797(0.043)** | 0.553(0.066) |
| | LDA | 0.275(0.028) | 0.695(0.047) | 0.755(0.040) | 0.452(0.056) |
| | QDA | **0.208(0.020)** | **0.827(0.026)** | 0.757(0.032) | **0.587(0.039)** |
| | SSQDA | 0.228(0.040) | 0.749(0.079) | 0.795(0.043) | 0.547(0.076) |
| 400 | SDAR | **0.235(0.024)** | **0.755(0.037)** | **0.775(0.032)** | **0.531(0.049)** |
| | SLDA | **0.235(0.024)** | **0.755(0.037)** | **0.775(0.032)** | **0.531(0.049)** |
| | LDA | 0.289(0.022) | 0.701(0.036) | 0.722(0.035) | 0.423(0.044) |
| | QDA | 0.335(0.024) | 0.692(0.035) | 0.639(0.037) | 0.332(0.048) |
| | SSQDA | 0.236(0.026) | 0.752(0.039) | 0.777(0.033) | 0.529(0.052) |

### A.2 EXTENSION TO MULTIGROUP CLASSIFICATION

We first extend the theory to unequal prior probabilities setting, where $\pi_1, \pi_2$ can be estimated by

$$\hat{\pi}_1 = \frac{n_1}{n_1 + n_2}, \quad \hat{\pi}_2 = \frac{n_2}{n_1 + n_2}.$$

The corresponding SSQDA rule can be written as

$$\widetilde{Q}(\boldsymbol{z}) = (\boldsymbol{z} - \tilde{\boldsymbol{\mu}}_1)^\top \tilde{\mathbf{D}} (\boldsymbol{z} - \tilde{\boldsymbol{\mu}}_1) - 2\tilde{\boldsymbol{\beta}}^\top (\boldsymbol{z} - \bar{\tilde{\boldsymbol{\mu}}})$$

$$- \log \left| \tilde{\mathbf{D}} \tilde{\boldsymbol{\Sigma}}_1 + \mathbf{I}_p \right| + \log \left( \frac{\hat{\pi}_1}{\hat{\pi}_2} \right).$$

As in Cai et al. (2011), the prior ratio converges fast:

$$\mathbb{P} \left( \left| 2 \log \left( \frac{\pi_1}{\pi_2} \right) - \log \left( \frac{\hat{\pi}_1}{\hat{\pi}_2} \right) \right| > M n^{-1/2} \right) \leq e^{-cM}.$$

Therefore, the convergence rate in Theorem 3.2 remains unchanged.

This idea and theory can also be extended to the classification problem involving multiple populations. Assume there are $K$ groups with distribution

$$EC_p(\boldsymbol{\mu}_k, \boldsymbol{\Sigma}_k, r), \quad k = 1, \ldots, K.$$

Table 9: Comparison of different methods under $t_5$ distribution under Model 3.

| p | | Error rate | Specificity | Sensitivity | Mcc |
|---|---|---|---|---|---|
| 100 | SDAR | 0.197(0.024) | 0.793(0.032) | 0.812(0.032) | 0.606(0.047) |
| | SLDA | 0.197(0.024) | 0.793(0.032) | 0.812(0.032) | 0.606(0.047) |
| | LDA | 0.237(0.026) | 0.753(0.041) | 0.774(0.036) | 0.527(0.052) |
| | QDA | 0.226(0.025) | **0.805(0.040)** | 0.744(0.044) | 0.551(0.050) |
| | SSQDA | **0.188(0.020)** | 0.804(0.032) | **0.820(0.028)** | **0.625(0.039)** |
| 200 | SDAR | 0.313(0.035) | 0.686(0.070) | 0.689(0.050) | 0.376(0.068) |
| | SLDA | 0.311(0.032) | 0.692(0.040) | 0.687(0.044) | 0.379(0.063) |
| | LDA | 0.369(0.029) | 0.605(0.050) | 0.657(0.048) | 0.263(0.058) |
| | QDA | 0.340(0.025) | **0.714(0.041)** | 0.607(0.043) | 0.323(0.050) |
| | SSQDA | **0.294(0.029)** | 0.707(0.040) | **0.705(0.042)** | **0.412(0.057)** |
| 400 | SDAR | 0.359(0.029) | 0.646(0.071) | 0.637(0.055) | 0.284(0.054) |
| | SLDA | 0.358(0.027) | 0.651(0.047) | 0.633(0.043) | 0.285(0.054) |
| | LDA | 0.396(0.027) | 0.550(0.049) | **0.659(0.039)** | 0.210(0.053) |
| | QDA | 0.433(0.025) | 0.639(0.046) | 0.495(0.044) | 0.137(0.051) |
| | SSQDA | **0.337(0.029)** | **0.668(0.040)** | 0.658(0.043) | **0.326(0.059)** |

Table 10: Comparison of different methods under mixture normal distribution under Model 3.

| p | | Error rate | Specificity | Sensitivity | Mcc |
|---|---|---|---|---|---|
| 100 | SDAR | 0.167(0.071) | 0.759(0.165) | 0.908(0.034) | 0.683(0.117) |
| | SLDA | 0.154(0.061) | 0.790(0.140) | 0.902(0.030) | 0.702(0.105) |
| | LDA | 0.144(0.027) | 0.817(0.050) | 0.896(0.022) | 0.715(0.052) |
| | QDA | **0.092(0.016)** | **0.882(0.026)** | **0.933(0.015)** | **0.817(0.032)** |
| | SSQDA | 0.107(0.035) | 0.875(0.076) | 0.911(0.021) | 0.789(0.062) |
| 200 | SDAR | 0.173(0.038) | 0.802(0.082) | 0.852(0.035) | 0.658(0.067) |
| | SLDA | 0.173(0.038) | 0.802(0.082) | 0.852(0.035) | 0.658(0.068) |
| | LDA | 0.194(0.023) | 0.759(0.044) | 0.853(0.027) | 0.616(0.045) |
| | QDA | 0.176(0.018) | 0.812(0.031) | 0.837(0.026) | 0.650(0.035) |
| | SSQDA | **0.146(0.052)** | **0.831(0.112)** | **0.877(0.028)** | **0.710(0.092)** |
| 400 | SDAR | 0.170(0.039) | 0.820(0.085) | 0.840(0.033) | 0.661(0.073) |
| | SLDA | 0.167(0.023) | 0.827(0.034) | 0.838(0.030) | 0.666(0.045) |
| | LDA | 0.199(0.023) | 0.759(0.038) | 0.842(0.027) | 0.603(0.045) |
| | QDA | 0.279(0.023) | 0.696(0.039) | 0.746(0.032) | 0.443(0.045) |
| | SSQDA | **0.143(0.040)** | **0.848(0.087)** | **0.866(0.026)** | **0.714(0.077)** |

Table 11: Performance of modern methods under normal distribution

| $p$ | Method | Error | Sensitivity | Specificity | MCC |
|---|---|---|---|---|---|
| 200 | Random Forest | **0.000(0.002)** | **1.000(0.002)** | **1.000(0.002)** | 1.000(0.003) |
| | Neural Net | 0.438(0.052) | 0.579(0.142) | 0.545(0.142) | 0.126(0.107) |
| | SVM | 0.334(0.049) | 0.672(0.057) | 0.661(0.074) | 0.334(0.099) |
| | Logistic | 0.361(0.046) | 0.740(0.047) | 0.538(0.072) | 0.284(0.093) |
| | KNN | 0.446(0.036) | 0.846(0.079) | 0.262(0.096) | 0.138(0.087) |
| | SSQDA | 0.228(0.031) | 0.767(0.042) | 0.777(0.045) | 0.482(0.092) |
| 400 | Random Forest | **0.001(0.003)** | **1.000(0.002)** | **0.998(0.006)** | **0.997(0.006)** |
| | Neural Net | 0.472(0.038) | 0.517(0.160) | 0.539(0.147) | 0.058(0.081) |
| | SVM | 0.377(0.047) | 0.621(0.077) | 0.625(0.066) | 0.247(0.094) |
| | Logistic | 0.385(0.041) | 0.699(0.055) | 0.532(0.067) | 0.235(0.083) |
| | KNN | 0.465(0.032) | 0.743(0.085) | 0.327(0.097) | 0.078(0.071) |
| | SSQDA | 0.281(0.034) | 0.717(0.043) | 0.721(0.043) | 0.439(0.067) |

We adopt the Bayesian classification criterion. A new observation $\boldsymbol{z}$ is assigned to class $k$ if and only if

$$k = \arg \min_{k \in \{1,\dots,K\}} Q_k(\boldsymbol{z}),$$

Table 12: Performance under $t$-distribution

| $p$ | Method | Error | Sensitivity | Specificity | MCC |
|---|---|---|---|---|---|
| | Random Forest | 0.352(0.050) | 0.655(0.078) | 0.641(0.075) | 0.299(0.100) |
| | Neural Net | 0.446(0.051) | 0.566(0.197) | 0.541(0.191) | 0.116(0.106) |
| 200 | SVM | 0.339(0.050) | 0.667(0.069) | 0.655(0.073) | 0.324(0.101) |
| | Logistic | 0.361(0.045) | 0.760(0.051) | 0.518(0.072) | 0.287(0.091) |
| | KNN | 0.454(0.042) | 0.510(0.249) | 0.581(0.253) | 0.107(0.087) |
| | SSQDA | **0.184(0.039)** | **0.816(0.075)** | **0.817(0.043)** | **0.634(0.074)** |
| | Random Forest | 0.346(0.039) | 0.639(0.071) | 0.669(0.059) | 0.309(0.079) |
| | Neural Net | 0.442(0.062) | 0.588(0.194) | 0.529(0.194) | 0.123(0.126) |
| 400 | SVM | 0.302(0.051) | 0.693(0.057) | 0.703(0.077) | 0.397(0.102) |
| | Logistic | 0.354(0.044) | 0.764(0.055) | 0.528(0.060) | 0.301(0.092) |
| | KNN | 0.430(0.046) | 0.528(0.207) | 0.611(0.199) | 0.149(0.094) |
| | SSQDA | **0.212(0.032)** | **0.791(0.040)** | **0.785(0.042)** | **0.577(0.063)** |

where

$$Q_k(\boldsymbol{z}) = \tfrac{1}{2}(\boldsymbol{z} - \boldsymbol{\mu}_k)^\top \mathbf{D}_k(\boldsymbol{z} - \boldsymbol{\mu}_k)$$
$$- \boldsymbol{\beta}_k^\top(\boldsymbol{z} - \bar{\boldsymbol{\mu}}_k) - \tfrac{1}{2}\log|\mathbf{D}_k\boldsymbol{\Sigma}_1 + \mathbf{I}_p| + \log\left(\tfrac{\pi_1}{\pi_k}\right), \quad k = 1, \ldots, K,$$

with

$$\mathbf{D}_k = \boldsymbol{\Omega}_k - \boldsymbol{\Omega}_1, \quad \bar{\boldsymbol{\mu}}_k = \tfrac{\boldsymbol{\mu}_1 + \boldsymbol{\mu}_k}{2}, \quad \boldsymbol{\beta}_k = \boldsymbol{\Omega}_1(\boldsymbol{\mu}_k - \boldsymbol{\mu}_1).$$

When the parameters are unknown, under sparsity assumptions on $\mathbf{D}_k$ and $\boldsymbol{\beta}_k$, we estimate them using samples $\boldsymbol{X}_1^{(k)}, \ldots, \boldsymbol{X}_{n_k}^{(k)} \sim EC_p(\boldsymbol{\mu}_k, \boldsymbol{\Sigma}_k, r)$:

$$\tilde{\mathbf{D}}_k = \arg\min_{\mathbf{D}\in\mathbb{R}^{p\times p}} \left\{ \|\text{vec}(\mathbf{D})\|_1 : \left\|\tfrac{1}{2}\tilde{\boldsymbol{\Sigma}}_1\mathbf{D}\tilde{\boldsymbol{\Sigma}}_k + \tfrac{1}{2}\tilde{\boldsymbol{\Sigma}}_k\mathbf{D}\tilde{\boldsymbol{\Sigma}}_1 - \tilde{\boldsymbol{\Sigma}}_1 + \tilde{\boldsymbol{\Sigma}}_k\right\|_{\max} \le \lambda_{1,n} \right\},$$

$$\tilde{\boldsymbol{\beta}}_k = \arg\min_{\boldsymbol{\beta}\in\mathbb{R}^p} \left\{ \|\boldsymbol{\beta}\|_1 : \|\tilde{\boldsymbol{\Sigma}}_1\boldsymbol{\beta} - \tilde{\boldsymbol{\mu}}_k + \tilde{\boldsymbol{\mu}}_1\|_\infty \le \lambda_{2,n} \right\},$$

where

$$\tilde{\boldsymbol{\Sigma}}_k = \widetilde{\text{tr}(\boldsymbol{\Sigma}_k)}\, \tilde{\mathbf{S}}_k,$$

$\tilde{\boldsymbol{\mu}}_k$ is the sample spatial median of group $k$, and $\lambda_{1,n}, \lambda_{2,n}$ are tuning parameters.

Therefore, the discriminating function is

$$\tilde{Q}_k(\boldsymbol{z}) = \tfrac{1}{2}(\boldsymbol{z} - \tilde{\boldsymbol{\mu}}_k)^\top \tilde{\mathbf{D}}_k(\boldsymbol{z} - \tilde{\boldsymbol{\mu}}_k)$$
$$- \tilde{\boldsymbol{\beta}}_k^\top\left(\boldsymbol{z} - \tfrac{\tilde{\boldsymbol{\mu}}_1 + \tilde{\boldsymbol{\mu}}_k}{2}\right)$$
$$- \tfrac{1}{2}\log\left|\tilde{\mathbf{D}}_k\tilde{\boldsymbol{\Sigma}}_1 + \mathbf{I}_p\right| + \log\left(\tfrac{\hat{\pi}_1}{\hat{\pi}_k}\right), \quad k = 1, \ldots, K.$$

Finally, the classification rule is

$$\tilde{G}_{\tilde{Q}}(\boldsymbol{z}) = \arg\min_{k\in\{1,\ldots,K\}} \tilde{Q}_k(\boldsymbol{z}).$$

The convergence rate for misclassification follows from the same techniques as in Theorem 3.2.

### A.3 PROOF OF THEOREMS

In this section, we prove Theorem 3.1, Theorem 3.2 and Theorem 3.3.

### A.3.1 USEFUL LEMMAS

We begin by presenting several lemmas that establish the consistency of spatial median and sample spatial sign covariance. Let $\boldsymbol{X} \sim EC_p(\boldsymbol{\mu}, \boldsymbol{\Sigma}_0, r)$, $\mathbb{E}(r^2) = p$, $\Lambda_0 = \frac{p}{\text{tr}(\boldsymbol{\Sigma}_0)}\boldsymbol{\Sigma}_0$. Spatial sign covariance matrix is denoted by $\mathbf{S}$. Sample spatial sign convariance matrix and spatial median is denoted by $\tilde{\mathbf{S}}$ and $\tilde{\boldsymbol{\mu}}$ respectively.

**Lemma A.1.** *Under Assumption 3.3, 3.4 and 3.5, $\|\mathbf{S}\|_{max} = O(p^{-1})$.*

**Lemma A.2.** *Under Assumption 3.3, 3.4 and 3.5, when $p$ is large enough, $\exists C_{c_0,\eta,T,M} > 0, s.t.$*

$$\|pS - \Lambda_0\|_{max} \leq \frac{C_{c_0,\eta,T,M}}{\sqrt{p}}.$$

**Lemma A.3.** *Under Assumption 3.6, for any $\alpha > 0$, with probability over $1 - \alpha$*

$$\|\widetilde{\mathbf{S}} - \mathbf{S}\|_{max} \leq C\left(\frac{8\lambda_1(\boldsymbol{\Sigma}_0)}{p\lambda_p(\boldsymbol{\Sigma}_0)} + \|\mathbf{S}\|_{max}\right)\sqrt{\frac{\log p + \log(1/\sqrt{\alpha/2})}{n}},$$

$$\|\tilde{\boldsymbol{\mu}} - \boldsymbol{\mu}\|_{max} \leq \frac{2\lambda_1(\boldsymbol{\Sigma}_0)}{\zeta_1 p\lambda_p(\boldsymbol{\Sigma}_0)}\sqrt{\frac{\log(p) + \log(2/\alpha)}{n}}.$$

*By Jenson's inequality, we have $\zeta_1 = \mathbb{E}\left(\frac{1}{\|\boldsymbol{X}-\boldsymbol{\mu}\|_2}\right) \geq \sqrt{\frac{1}{\mathbb{E}(\|\boldsymbol{X}-\boldsymbol{\mu}\|_2^2)}} \geq \sqrt{\frac{1}{\mathbb{E}(r^2)\|\boldsymbol{\Sigma}_0\|_2}}$. Therefore, we can further obtain*

$$\|\tilde{\boldsymbol{\mu}} - \boldsymbol{\mu}\|_{max} \leq \frac{2\lambda_1(\boldsymbol{\Sigma}_0)}{\sqrt{p}\lambda_p(\boldsymbol{\Sigma}_0)}\sqrt{\frac{\log(p) + \log(2/\alpha)}{n}}.$$

*Proof.* Lemmas A.1, A.2 and A.3 are Lemmas $5, 6, 7$ in Section 5 of Lu & Feng (2025). $\square$

**Lemma A.4.** *If $Var(r^2) \lesssim p^2$,*

$$P\left(\left|\frac{\widetilde{tr(\boldsymbol{\Sigma}_0)}}{tr(\boldsymbol{\Sigma}_0)} - 1\right| \geq t\right) \lesssim \frac{1}{nt^2}.$$

*Proof.* Since

$$\widetilde{\mathrm{tr}(\boldsymbol{\Sigma}_0)} = \frac{\sum_{i\neq j\neq k}(\boldsymbol{X}_i - \boldsymbol{\mu} + \boldsymbol{\mu} - \boldsymbol{X}_j)^T(\boldsymbol{X}_k - \boldsymbol{\mu} + \boldsymbol{\mu} - \boldsymbol{X}_j)}{n(n-1)(n-2)},$$

without loss of generality, we can assume $\boldsymbol{\mu} = \boldsymbol{0}$. In the meantime,

$$\begin{aligned}
\mathbb{E}\left((\boldsymbol{X}_i - \boldsymbol{X}_j)^T(\boldsymbol{X}_k - \boldsymbol{X}_j)\right) =& \mathbb{E}\left(\boldsymbol{X}_i^T\boldsymbol{X}_k - \boldsymbol{X}_i^T\boldsymbol{X}_j - \boldsymbol{X}_j^T\boldsymbol{X}_k + \boldsymbol{X}_j^T\boldsymbol{X}_j\right) \\
=& \mathbb{E}(\boldsymbol{X}_j^T\boldsymbol{X}_j) = \mathrm{tr}(\boldsymbol{\Sigma}_0).
\end{aligned}$$

It is straightforward to show that $\widetilde{\mathrm{tr}(\boldsymbol{\Sigma}_0)}$ is an unbiased estimator. Next we consider $\mathrm{Var}(\widetilde{\mathrm{tr}(\boldsymbol{\Sigma}_0)})$. For $i \neq k \neq j$,

$$\begin{aligned}
\mathbb{E}(\boldsymbol{X}_i^T\boldsymbol{X}_k)^2 =& \mathbb{E}\left(\mathbb{E}(\boldsymbol{X}_i^T\boldsymbol{X}_k\boldsymbol{X}_k^T\boldsymbol{X}_i|\boldsymbol{X}_k)\right) \\
=& \mathbb{E}\left(\mathrm{tr}(\boldsymbol{X}_k\boldsymbol{X}_k^T\boldsymbol{\Sigma}_0)\right) \\
=& \mathrm{tr}(\boldsymbol{\Sigma}_0^2),
\end{aligned}$$

$$\begin{aligned}
\mathbb{E}(\boldsymbol{X}_i^T\boldsymbol{X}_i)^2 =& \mathbb{E}\left(r^4\right)\mathbb{E}\left(\boldsymbol{u}^T\boldsymbol{\Sigma}_0\boldsymbol{u}\right)^2 \\
=& \left[\mathrm{Var}(r^2) + \left(\mathbb{E}(r^2)\right)^2\right]\mathbb{E}\left(\boldsymbol{u}^T\boldsymbol{\Sigma}_0\boldsymbol{u}\right)^2 \\
\asymp& p^2\mathbb{E}\left(\boldsymbol{u}^T\boldsymbol{\Sigma}_0\boldsymbol{u}\right)^2,
\end{aligned}$$

where $\boldsymbol{u} = (u_1, u_2, \cdots, u_p)$ is uniformly distributed on $\mathbb{S}^{p-1}$. A well known result is that $\mathbb{E}(u_i^4) \asymp \mathbb{E}(u_i^2 u_j^2) \asymp \frac{1}{p^2}$. Let $\boldsymbol{\Sigma}_0 = (\sigma_{ij})_{p \times p}$, then

$$
\begin{aligned}
\mathbb{E}(\boldsymbol{u}^T \boldsymbol{\Sigma}_0 \boldsymbol{u})^2 =& \mathbb{E}\left( \sum_{i,j} \sigma_{ij} u_i u_j \right)^2 \\
=& \mathbb{E}\left( \sum_{i,j,l,m} \sigma_{ij} \sigma_{lm} u_i u_j u_l u_m \right) \\
=& \sum_{i,l} \sigma_{ii} \sigma_{ll} \mathbb{E}\left( u_i^2 u_l^2 \right) + \sum_{i,j} \sigma_{i,j}^2 \mathbb{E}\left( u_i^2 u_j^2 \right) + \sum_i \sigma_{ii}^2 \mathbb{E}\left( u_i^4 \right) \\
\asymp& \frac{(\mathrm{tr}(\boldsymbol{\Sigma}_0))^2 + \mathrm{tr}(\boldsymbol{\Sigma}_0{}^2)}{p^2}.
\end{aligned}
$$

Thus, $\mathbb{E}(\boldsymbol{X}_i{}^T \boldsymbol{X}_i)^2 \asymp (\mathrm{tr}(\boldsymbol{\Sigma}_0))^2 + \mathrm{tr}(\boldsymbol{\Sigma}_0{}^2)$. For other combinations of quadratic terms, the expectation is 0. Therefore, by considering the possible combinations, we can obtain

$$
\begin{aligned}
\mathrm{Var}(\widetilde{\mathrm{tr}(\boldsymbol{\Sigma}_0)}) =& \frac{\mathbb{E}\left( \sum_{i \neq j \neq k}(\boldsymbol{X}_i - \boldsymbol{X}_j)^T(\boldsymbol{X}_k - \boldsymbol{X}_j) \sum_{l \neq m \neq n}(\boldsymbol{X}_l - \boldsymbol{X}_m)^T(\boldsymbol{X}_n - \boldsymbol{X}_m) \right)}{n^2(n-1)^2(n-2)^2} \\
& - (\mathrm{tr}(\boldsymbol{\Sigma}_0))^2 \\
=& \frac{n^6 + O(n^5)}{n^2(n-1)^2(n-2)^2} (\mathrm{tr}(\boldsymbol{\Sigma}_0))^2 + O\left(\frac{1}{n}\right) \left[ \mathrm{tr}(\boldsymbol{\Sigma}_0{}^2) + (\mathrm{tr}(\boldsymbol{\Sigma}_0))^2 \right] - (\mathrm{tr}(\boldsymbol{\Sigma}_0))^2 \\
\leq& O\left(\frac{1}{n}\right)(\mathrm{tr}(\boldsymbol{\Sigma}_0))^2.
\end{aligned}
$$

Lastly, by chebyshelve's inequality:

$$
P\left( \left| \frac{\widetilde{\mathrm{tr}(\boldsymbol{\Sigma}_0)}}{\mathrm{tr}(\boldsymbol{\Sigma}_0)} - 1 \right| \geq t \right) \leq \frac{\mathrm{Var}\left( \frac{\widetilde{\mathrm{tr}(\boldsymbol{\Sigma}_0)}}{\mathrm{tr}(\boldsymbol{\Sigma}_0)} \right)}{t^2} \lesssim \frac{1}{nt^2}.
$$

$\square$

**Lemma A.5.** *With probability over* $1 - O\left(\frac{1}{\log p}\right)$, *we have*

$$
\|\tilde{\boldsymbol{\Sigma}}_0 - \boldsymbol{\Sigma}_0\|_{max} \lesssim \sqrt{\frac{1}{p}} + \sqrt{\frac{\log(p)}{n}}.
$$

*Proof.* By Lemma A.1 and Assumtion 3.4, we have

$$
\frac{8\lambda_1(\boldsymbol{\Sigma}_0)}{p\lambda_p(\boldsymbol{\Sigma}_0)} + \|\mathbf{S}\|_{\max} \lesssim \frac{1}{p}.
$$

By Assumption 3.5, $\mathrm{tr}(\boldsymbol{\Sigma}_0) \asymp p$. Let $t = \sqrt{\frac{\log p}{n}}$ in Lemma A.4, by Lemma A.2, and triangle inequality, we obtain

$$
\begin{aligned}
\|\tilde{\boldsymbol{\Sigma}}_0 - \boldsymbol{\Sigma}_0\|_{\max} \leq& \frac{\mathrm{tr}(\boldsymbol{\Sigma}_0)}{p} \left( \left\| \left( \frac{\widetilde{\mathrm{tr}(\boldsymbol{\Sigma}_0)}}{\mathrm{tr}\boldsymbol{\Sigma}_0} - 1 \right) p\tilde{\mathbf{S}} \right\|_{\max} + \|p\tilde{\mathbf{S}} - \boldsymbol{\Lambda}_0\|_{\max} \right) \\
\lesssim& \sqrt{\frac{1}{p}} + \sqrt{\frac{\log(p)}{n}},
\end{aligned}
$$

with probability over $1 - O\left(\frac{1}{\log p}\right)$. $\square$

**Lemma A.6.** *When $(s\log(ep/s) + \log(1/\alpha))/n \to 0$, with probability at least $1 - 3\alpha$, we have*

$$\|\widetilde{\mathbf{S}} - \mathbf{S}\|_{2,s} \leq C_0 \left( \sup_{\boldsymbol{v} \in \mathbb{S}^{p-1}} 2\|\boldsymbol{v}^T U(\boldsymbol{X} - \boldsymbol{\mu})\|_{\psi_2}^2 + \|\mathbf{S}\|_2 \right) \sqrt{\frac{s(3 + \log(p/s)) + \log(1/\alpha)}{n}}$$

$$+ C_1 \left( \frac{np}{s} \right)^{-\frac{1}{2}(1+\delta)},$$

*for some absolute constants $C_0, C_1 > 0$ and $\delta \in (0, 1)$.*

*Proof.* This is Theorem 3.1 of Feng (2024). $\qquad\square$

**Remark A.1.** *As a special case within Theorem 4.2 in Han & Liu (2018), when $\frac{\lambda_1(\boldsymbol{\Sigma}_0)}{\lambda_p(\boldsymbol{\Sigma}_0)}$ is upper bounded by some positive constant, we have*

$$\sup_{\boldsymbol{v}} \|\boldsymbol{v}^T S(\boldsymbol{X})\|_{\psi_2}^2 \asymp \frac{1}{p},$$

*where $S(\cdot)$ is the self-normalized operator, with the fact that $U(\boldsymbol{X}) \stackrel{d}{=} S(\boldsymbol{X})$ when $\boldsymbol{X}$ follows elliptical distribution with mean 0. Therefore, we have*

$$\|\widetilde{\mathbf{S}} - \mathbf{S}\|_{2,s} \lesssim \left( \frac{1}{p} + \|\mathbf{S}\|_2 \right) \sqrt{\frac{s(3 + \log(p/s)) + \log(1/\alpha)}{n}} + C_1 \left( \frac{np}{s} \right)^{-\frac{1}{2}(1+\delta)}. \qquad (10)$$

**Lemma A.7.** *Under Assumption 3.4, we have*

$$\|\mathbf{S}\|_2 \lesssim \frac{1}{p}.$$

*Proof.* By Feng (2024), we obtain the relationship between the eigenvalues of the spatial sign covariance $\mathbf{S}$ and $\boldsymbol{\Sigma}_0$

$$\lambda_j(\mathbf{S}) = \mathbb{E} \left( \frac{\lambda_j(\boldsymbol{\Sigma}_0) \boldsymbol{Y}_j^2}{\lambda_1(\boldsymbol{\Sigma}_0) \boldsymbol{Y}_1^2 + \cdots + \lambda_p(\boldsymbol{\Sigma}_0) \boldsymbol{Y}_p^2} \right),$$

*where $\boldsymbol{Y}_1, \boldsymbol{Y}_2, \cdots, \boldsymbol{Y}_p \stackrel{i.i.d.}{\sim} N(0, 1)$. Since $\lambda_i(\boldsymbol{\Sigma}_0)$ are lower bounded by a positive constant $M_1^{-1}$,*

$$\lambda_j(\mathbf{S}) \leq M_1 \lambda_j(\boldsymbol{\Sigma}_0) \mathbb{E} \left( \frac{\boldsymbol{Y}_j^2}{\boldsymbol{Y}_1^2 + \cdots + \boldsymbol{Y}_p^2} \right)$$

$$\lesssim \mathbb{E} \left( \frac{\boldsymbol{Y}_j^2}{\boldsymbol{Y}_1^2 + \cdots + \boldsymbol{Y}_p^2} \right).$$

As is mentioned in Proposition 2.1 of Han & Liu (2018), $\frac{\boldsymbol{Y}_j^2}{\boldsymbol{Y}_1^2 + \cdots + \boldsymbol{Y}_p^2} \sim \text{Beta}(\frac{1}{2}, \frac{p-1}{2})$ with mean $\frac{1}{p}$, we reach the conclusion. $\qquad\square$

The following are technical lemmas.

**Lemma A.8.** *Suppose $\boldsymbol{x}, \boldsymbol{y} \in \mathbb{R}^p$. Let $\boldsymbol{h} = \boldsymbol{x} - \boldsymbol{y}$. Denote $\mathcal{S} = \text{supp}(\boldsymbol{y})$ and $s = |\mathcal{S}|$. If $\|\boldsymbol{x}\|_1 \leq \|\boldsymbol{y}\|_1$, then $\boldsymbol{h} \in \Gamma_1(s; p)$, that is,*

$$\|\boldsymbol{h}_{\mathcal{S}^c}\|_1 \leq \|\boldsymbol{h}_{\mathcal{S}}\|_1.$$

**Lemma A.9.** *For positive matrix $\boldsymbol{X}, \boldsymbol{Y}$,*

$$\log |\boldsymbol{X}| \leq \log |\boldsymbol{Y}| + tr(Y^{-1}(\boldsymbol{X} - \boldsymbol{Y})).$$

**Lemma A.10.** *(Von Neumann Lemma) Let $\mathbf{E} \in \mathbb{R}^{p \times p}$ with $\|\mathbf{E}\| < 1$, where $\|\cdot\|$ is a consistent matrix norm satisfying $\|\mathbf{I}_p\| = 1$, then $\mathbf{I}_p - \mathbf{E}$ is invertible and*

$$\|(\mathbf{I}_p - \mathbf{E})^{-1}\| \leq \frac{1}{1 - \|\mathbf{E}\|}.$$

**Lemma A.11.** *For a symmetric matrix* $\mathbf{M} = (m_{ij})_{p \times p}$ *and a positive constant* $s$ ,

$$\|\mathbf{M}\|_{2,s} \leq s\|\mathbf{M}\|_{max}.$$

*Proof.* Let $\boldsymbol{v} = (v_i)_{i=1\cdots p}$ be a vector with $\|\boldsymbol{v}\|_2 = 1, \|\boldsymbol{v}\|_0 \leq s$ . Without loss of generality, assume its non-zero elements are among $v_{k_1} \cdots v_{k_s}$. We have

$$\boldsymbol{v}^T \mathbf{M} \boldsymbol{v} = \sum_{i=1}^p \sum_{j=1}^p v_i m_{ij} v_j = \sum_{i=k_1}^{k_s} \sum_{j=k_1}^{k_s} v_i m_{ij} v_j$$

$$\leq \|\mathbf{M}\|_{\max} \|\boldsymbol{v}\|_1^2 \leq s\|\mathbf{M}\|_{\max}.$$

$\square$

### A.3.2 THE PROOF OF THEOREM 3.1

**Lemma A.12.** *With probability* $1 - O\left(\frac{1}{\log p}\right)$, *we have*

$$Vec(\tilde{\mathbf{D}} - \mathbf{D}) \in \Gamma_1(s_1; p^2).$$

*Proof.* By Lemma A.8. It suffices to prove $\|Vec(\tilde{\mathbf{D}})\|_1 \leq \|Vec(\mathbf{D})\|_1$ which follows directly from the fact that $\mathbf{D}$ is a feasible solution to problem (3). Denote $\tilde{\boldsymbol{\Sigma}}_i = \widetilde{\mathrm{tr}(\boldsymbol{\Sigma}_i)}\tilde{\mathbf{S}}_{\mathbf{i}}, \mathbf{V} = \frac{1}{2}\boldsymbol{\Sigma}_1 \otimes \boldsymbol{\Sigma}_2 + \frac{1}{2}\boldsymbol{\Sigma}_2 \otimes \boldsymbol{\Sigma}_1, \boldsymbol{v_\Sigma} = \mathrm{vec}(\boldsymbol{\Sigma}_1) - \mathrm{vec}(\boldsymbol{\Sigma}_2)$ and $\tilde{\mathbf{V}} = \frac{1}{2}\tilde{\boldsymbol{\Sigma}}_1 \otimes \tilde{\boldsymbol{\Sigma}}_2 + \frac{1}{2}\tilde{\boldsymbol{\Sigma}}_2 \otimes \tilde{\boldsymbol{\Sigma}}_1, \widetilde{\boldsymbol{v_\Sigma}} = \mathrm{vec}(\tilde{\boldsymbol{\Sigma}}_1) - \mathrm{vec}(\tilde{\boldsymbol{\Sigma}}_2)$. Observe that

$$\mathbf{V} Vec(\mathbf{D}) = \boldsymbol{v_\Sigma}.$$

Thus,

$$\begin{aligned}
\|\tilde{\mathbf{V}} Vec(\mathbf{D}) - \widetilde{\boldsymbol{v_\Sigma}}\|_\infty &= \|\tilde{\mathbf{V}} Vec(\mathbf{D}) - \mathbf{V} Vec(\mathbf{D}) + \boldsymbol{v_\Sigma} - \widetilde{\boldsymbol{v_\Sigma}}\|_\infty \\
&\leq \|(\tilde{\mathbf{V}} - \mathbf{V}) Vec(\mathbf{D})\|_\infty + \|\boldsymbol{v_\Sigma} - \widetilde{\boldsymbol{v_\Sigma}}\|_\infty \\
&\leq \|(\tilde{\mathbf{V}} - \mathbf{V}) Vec(\mathbf{D})\|_\infty + \|Vec(\boldsymbol{\Sigma}_1) - Vec(\tilde{\boldsymbol{\Sigma}}_1)\|_\infty \\
&\quad + \|Vec(\boldsymbol{\Sigma}_2) - Vec(\tilde{\boldsymbol{\Sigma}}_2)\|_\infty \\
&\leq \|\tilde{\mathbf{V}} - \mathbf{V}\|_{\max} \|Vec(\mathbf{D})\|_1 + \|Vec(\boldsymbol{\Sigma}_1) - Vec(\tilde{\boldsymbol{\Sigma}}_1)\|_\infty \\
&\quad + \|Vec(\boldsymbol{\Sigma}_2) - Vec(\tilde{\boldsymbol{\Sigma}}_2)\|_\infty \\
&\leq \|\tilde{\mathbf{V}} - \mathbf{V}\|_{\max} \sqrt{s_1} M_0 + \|Vec(\boldsymbol{\Sigma}_1) - Vec(\tilde{\boldsymbol{\Sigma}}_1)\|_\infty \\
&\quad + \|Vec(\boldsymbol{\Sigma}_2) - Vec(\tilde{\boldsymbol{\Sigma}}_2)\|_\infty.
\end{aligned}$$

By Lemma A.5 , we have $\|Vec(\boldsymbol{\Sigma}_i) - Vec(\tilde{\boldsymbol{\Sigma}}_i)\|_\infty = \|\boldsymbol{\Sigma}_i - \tilde{\boldsymbol{\Sigma}}_i\|_{\max} \lesssim \sqrt{\frac{1}{p}} + \sqrt{\frac{\log(p)}{n}}$ with probability over $1 - O\left(\frac{1}{\log p}\right)$ . As for the first term

$$\|\tilde{\mathbf{V}} - \mathbf{V}\|_{\max} \leq \frac{1}{2}\|\tilde{\boldsymbol{\Sigma}}_1 \otimes \tilde{\boldsymbol{\Sigma}}_2 - \boldsymbol{\Sigma}_1 \otimes \boldsymbol{\Sigma}_2\|_{\max} + \frac{1}{2}\|\tilde{\boldsymbol{\Sigma}}_2 \otimes \tilde{\boldsymbol{\Sigma}}_1 - \boldsymbol{\Sigma}_2 \otimes \boldsymbol{\Sigma}_1\|_{\max}.$$

It suffices to consider

$$\|\tilde{\boldsymbol{\Sigma}}_1 \otimes \tilde{\boldsymbol{\Sigma}}_2 - \boldsymbol{\Sigma}_1 \otimes \boldsymbol{\Sigma}_2\|_{\max} \leq \|\tilde{\boldsymbol{\Sigma}}_1 \otimes (\tilde{\boldsymbol{\Sigma}}_2 - \boldsymbol{\Sigma}_2)\|_{\max} + \|(\tilde{\boldsymbol{\Sigma}}_1 - \boldsymbol{\Sigma}_1) \otimes \tilde{\boldsymbol{\Sigma}}_2\|_{\max}.$$

Since

$$\begin{aligned}
\|\tilde{\boldsymbol{\Sigma}}_1 \otimes (\tilde{\boldsymbol{\Sigma}}_2 - \boldsymbol{\Sigma}_2)\|_{\max} &\leq \|\tilde{\boldsymbol{\Sigma}}_1\|_{\max} \|\tilde{\boldsymbol{\Sigma}}_2 - \boldsymbol{\Sigma}_2\|_{\max} \\
&\leq (\|\boldsymbol{\Sigma}_1\|_{\max} + \|\tilde{\boldsymbol{\Sigma}}_1 - \boldsymbol{\Sigma}_1\|_{\max}) \|\tilde{\boldsymbol{\Sigma}}_2 - \boldsymbol{\Sigma}_2\|_{\max},
\end{aligned}$$

$\|\tilde{\mathbf{V}} - \mathbf{V}\|_{\max} \lesssim \|\tilde{\boldsymbol{\Sigma}}_2 - \boldsymbol{\Sigma}_2\|_{\max} \lesssim \sqrt{\frac{1}{p}} + \sqrt{\frac{\log(p)}{n}}$. Therefore with $\lambda_{1,n} = c_1 \sqrt{s_1} \left(\sqrt{\frac{1}{p}} + \sqrt{\frac{\log(p)}{n}}\right)$, where $c_1$ is a large enough constant, we have $\|\tilde{\mathbf{V}} Vec(\mathbf{D}) - \widetilde{\boldsymbol{v_\Sigma}}\|_\infty \leq \lambda_{1,n}$, which leads to $Vec(\tilde{\mathbf{D}} - \mathbf{D}) \in \Gamma_1(s_1; p^2)$. $\square$

Through simple computations, we obtain a straightforward corollary derived from Lemma A.12 :

$$\|\text{Vec}(\tilde{\mathbf{D}} - \text{Vec}(\mathbf{D}))\|_1 \le 2\sqrt{s_1}\|\text{Vec}(\tilde{\mathbf{D}} - \text{Vec}(\mathbf{D}))\|_2. \tag{11}$$

Through an entirely analogous process, with $\lambda_{2,n} = c_2\sqrt{s_2}\left(\sqrt{\frac{1}{p}} + \sqrt{\frac{\log(p)}{n}}\right)$, the following inequality also holds with probability of $1 - O\left(\frac{1}{\log p}\right)$,

$$\|\tilde{\boldsymbol{\beta}} - \boldsymbol{\beta}\|_1 \le 2\sqrt{s_2}\|\tilde{\boldsymbol{\beta}} - \boldsymbol{\beta}\|_2.$$

*Proof.* With $\left\|\mathbf{V}^{-1}\right\|_2 = \|\boldsymbol{\Omega}_1 \otimes \boldsymbol{\Omega}_2\|_2 = \|\boldsymbol{\Omega}_1\|_2 \cdot \|\boldsymbol{\Omega}_2\|_2 \le M_1^2$, consider $\|\mathbf{D} - \tilde{\mathbf{D}}\|_F$.

$$
\begin{aligned}
\|\mathbf{D} - \tilde{\mathbf{D}}\|_F^2 =& \|\text{vec}(\hat{\mathbf{D}}) - \text{vec}(\mathbf{D})\|_2^2 \le M_1^2 \lambda_{min}(\mathbf{V})\|\text{vec}(\hat{\mathbf{D}}) - \text{vec}(\mathbf{D})\|_2^2 \\
\le& M_1^2 |(Vec(\tilde{\mathbf{D}}) - Vec(\mathbf{D}))^T V (Vec(\tilde{\mathbf{D}}) - Vec(\mathbf{D}))| \\
\lesssim& |(Vec(\tilde{\mathbf{D}}) - Vec(\mathbf{D}))^T V (Vec(\tilde{\mathbf{D}}) - Vec(\mathbf{D}))| \\
=& |(Vec(\tilde{\mathbf{D}}) - Vec(\mathbf{D}))^T (V Vec(\tilde{\mathbf{D}}) - v_\Sigma)| \\
=& |(Vec(\tilde{\mathbf{D}}) - Vec(\mathbf{D}))^T ((\mathbf{V} - \tilde{\mathbf{V}})Vec(\tilde{\mathbf{D}}) + (\tilde{\mathbf{V}}Vec(\tilde{\mathbf{D}})) - \widetilde{v_\Sigma}) + (\widetilde{v_\Sigma} - v_\Sigma))| \\
\le& |(Vec(\tilde{\mathbf{D}}) - Vec(\mathbf{D}))^T (\mathbf{V} - \tilde{\mathbf{V}})Vec(\tilde{\mathbf{D}})| \\
& + |(Vec(\tilde{\mathbf{D}}) - Vec(\mathbf{D}))^T (\tilde{\mathbf{V}}Vec(\tilde{\mathbf{D}})) - \widetilde{v_\Sigma})| \\
& + |(Vec(\tilde{\mathbf{D}}) - Vec(\mathbf{D}))^T (\widetilde{v_\Sigma} - v_\Sigma))|.
\end{aligned}
$$

By triangle inequality,

$$
\begin{aligned}
\|\mathbf{D} - \tilde{\mathbf{D}}\|_F^2 \le& \|(Vec(\tilde{\mathbf{D}}) - Vec(\mathbf{D}))\|_1 \|(\mathbf{V} - \tilde{\mathbf{V}})Vec(\tilde{\mathbf{D}})\|_\infty \\
& + \|(Vec(\tilde{\mathbf{D}}) - Vec(\mathbf{D}))\|_1 \|(\tilde{\mathbf{V}}Vec(\tilde{\mathbf{D}})) - \widetilde{v_\Sigma}\|_\infty \\
& + \|(Vec(\tilde{\mathbf{D}}) - Vec(\mathbf{D}))\|_1 \|(\widetilde{v_\Sigma} - v_\Sigma))\|_\infty \\
\lesssim& \|(Vec(\tilde{\mathbf{D}}) - Vec(\mathbf{D}))\|_2 \sqrt{s_1}\|(\mathbf{V} - \tilde{\mathbf{V}})Vec(\tilde{\mathbf{D}})\|_\infty \\
& + \|(Vec(\tilde{\mathbf{D}}) - Vec(\mathbf{D}))\|_2 \sqrt{s_1}\|(\tilde{\mathbf{V}}Vec(\tilde{\mathbf{D}})) - \widetilde{v_\Sigma}\|_\infty \\
& + \|(Vec(\tilde{\mathbf{D}}) - Vec(\mathbf{D}))\|_2 \sqrt{s_1}\|(\widetilde{v_\Sigma} - v_\Sigma)\|_\infty.
\end{aligned}
$$

The last inequality uses (11). For the last two terms,

$$\|(\tilde{\mathbf{V}}Vec(\tilde{\mathbf{D}})) - \widetilde{v_\Sigma})\|_\infty \le \lambda_{1,n} \lesssim \sqrt{s_1}\left(\sqrt{\frac{1}{p}} + \sqrt{\frac{\log(p)}{n}}\right),$$

$$\|(\widetilde{v_\Sigma} - v_\Sigma))\|_\infty \le \|Vec(\boldsymbol{\Sigma}_1) - Vec(\tilde{\boldsymbol{\Sigma}}_1)\|_\infty + \|Vec(\boldsymbol{\Sigma}_2) - Vec(\tilde{\boldsymbol{\Sigma}}_2)\|_\infty \lesssim \left(\sqrt{\frac{1}{p}} + \sqrt{\frac{\log(p)}{n}}\right).$$

For the first term,

$$
\begin{aligned}
\|(\mathbf{V} - \tilde{\mathbf{V}})Vec(\tilde{\mathbf{D}})\|_\infty \le& \|(\mathbf{V} - \tilde{\mathbf{V}})(Vec(\tilde{\mathbf{D}}) - Vec(\mathbf{D}))\|_\infty + \|(\tilde{\mathbf{V}} - \mathbf{V})Vec(\mathbf{D})\|_\infty \\
\le& \|(\mathbf{V} - \tilde{\mathbf{V}})\|_\infty \|(Vec(\tilde{\mathbf{D}}) - Vec(\mathbf{D}))\|_1 + \|(\tilde{\mathbf{V}} - \mathbf{V})\|_{max}\sqrt{s_1}M_0 \\
\le& \sqrt{s_1}\left(\sqrt{\frac{1}{p}} + \sqrt{\frac{\log(p)}{n}}\right)\|Vec(\tilde{\mathbf{D}}) - Vec(\mathbf{D}))\|_2 \\
& + \sqrt{s_1}\left(\sqrt{\frac{1}{p}} + \sqrt{\frac{\log(p)}{n}}\right)M_0.
\end{aligned}
$$

Therefore $\|\tilde{\mathbf{D}} - \mathbf{D}\|_F \lesssim s_1\left(\sqrt{\frac{1}{p}} + \sqrt{\frac{\log(p)}{n}}\right)$ with probability over $1 - O\left(\frac{1}{\log p}\right)$.

The proof of $\|\tilde{\boldsymbol{\beta}} - \boldsymbol{\beta}\|_2$ follows the same process. By Assumption 3.4, we have

$$\|\boldsymbol{\beta} - \tilde{\boldsymbol{\beta}}\|_2^2 \lesssim |(\boldsymbol{\beta} - \tilde{\boldsymbol{\beta}})^T \boldsymbol{\Sigma}_2 (\boldsymbol{\beta} - \tilde{\boldsymbol{\beta}})|$$

$$\leq \left| (\tilde{\boldsymbol{\beta}} - \boldsymbol{\beta})^\top (\tilde{\boldsymbol{\Sigma}}_2 \tilde{\boldsymbol{\beta}} - \tilde{\boldsymbol{\delta}}) \right| + \left| (\tilde{\boldsymbol{\beta}} - \boldsymbol{\beta})^\top (\tilde{\boldsymbol{\Sigma}}_2 - \boldsymbol{\Sigma}_2) \tilde{\boldsymbol{\beta}} \right| + \left| (\tilde{\boldsymbol{\beta}} - \boldsymbol{\beta})^\top (\boldsymbol{\delta} - \tilde{\boldsymbol{\delta}}) \right|$$

$$\leq \sqrt{s_2} \|\tilde{\boldsymbol{\beta}} - \boldsymbol{\beta}\|_2 (\|\tilde{\boldsymbol{\Sigma}}_2 \tilde{\boldsymbol{\beta}} - \tilde{\boldsymbol{\delta}}\|_\infty + \|(\tilde{\boldsymbol{\Sigma}}_2 - \boldsymbol{\Sigma}_2) \tilde{\boldsymbol{\beta}}\|_\infty + \|\boldsymbol{\delta} - \tilde{\boldsymbol{\delta}}\|_\infty)$$

$$\leq \sqrt{s_2} \|\tilde{\boldsymbol{\beta}} - \boldsymbol{\beta}\|_2 \left( \lambda_{2,n} + \|(\tilde{\boldsymbol{\Sigma}}_2 - \boldsymbol{\Sigma}_2) \boldsymbol{\beta}\|_\infty + \|(\tilde{\boldsymbol{\Sigma}}_2 - \boldsymbol{\Sigma}_2)(\tilde{\boldsymbol{\beta}} - \boldsymbol{\beta})\|_\infty + \|\boldsymbol{\delta} - \tilde{\boldsymbol{\delta}}\|_\infty \right).$$

By Lemma A.3, $\|\boldsymbol{\delta} - \tilde{\boldsymbol{\delta}}\|_\infty \lesssim \sqrt{\frac{\log p}{n}}$ Thus $\|\boldsymbol{\beta} - \tilde{\boldsymbol{\beta}}\|_2 \lesssim s_2 \left( \sqrt{\frac{1}{p}} + \sqrt{\frac{\log(p)}{n}} \right)$, with probability of $1 - O\left( \frac{1}{\log p} \right)$. $\qquad\square$

### A.3.3 THE PROOF OF THEOREM 3.2

*Proof.* Given $\pi_2 = \pi_1 = \frac{1}{2}$, we first simplify the excess risk $R(G_{\widetilde{Q}}) - R(G_Q)$.

$$R(G_{\widetilde{Q}}) - R(G_Q) = \left( \pi_1 - \int_{\widetilde{Q}(\boldsymbol{z})>0} \pi_1 f_1(\boldsymbol{z})\, dz + \pi_2 - \int_{\widetilde{Q}(\boldsymbol{z})\leq 0} \pi_2 f_2(\boldsymbol{z})\, dz \right) -$$

$$\left( \pi_1 - \int_{Q(\boldsymbol{z})>0} \pi_1 f_1(\boldsymbol{z})\, dz + \pi_2 - \int_{Q(\boldsymbol{z})\leq 0} \pi_2 f_2(\boldsymbol{z})\, dz \right)$$

$$= \int_{Q(\boldsymbol{z})>0} \pi_1 f_1(\boldsymbol{z})\, dz + \int_{Q(\boldsymbol{z})\leq 0} \pi_2 f_2(\boldsymbol{z})\, dz$$

$$- \int_{\widetilde{Q}(\boldsymbol{z})>0} \pi_1 f_1(\boldsymbol{z})\, dz - \int_{\widetilde{Q}(\boldsymbol{z})\leq 0} \pi_2 f_2(\boldsymbol{z})\, dz$$

$$= \int_{Q(\boldsymbol{z})>0} \pi_1 f_1(\boldsymbol{z})\, dz + 1 - \int_{Q(\boldsymbol{z})>0} \pi_2 f_2(\boldsymbol{z})\, dz$$

$$- \int_{\widetilde{Q}(\boldsymbol{z})>0} \pi_1 f_1(\boldsymbol{z})\, dz - 1 + \int_{\widetilde{Q}(\boldsymbol{z})>0} \pi_2 f_2(\boldsymbol{z})\, dz$$

$$= \int_{Q(\boldsymbol{z})>0} \pi_1 f_1(\boldsymbol{z}) - \pi_2 f_2(\boldsymbol{z})\, dz - \int_{\widetilde{Q}(\boldsymbol{z})>0} \pi_1 f_1(\boldsymbol{z}) - \pi_2 f_2(\boldsymbol{z})\, dz$$

$$= \int_{Q(\boldsymbol{z})>0} \pi_1 f_1(\boldsymbol{z}) - \pi_2 f_2(\boldsymbol{z})\, dz - \int_{\widetilde{Q}(\boldsymbol{z})>0} \pi_1 f_1(\boldsymbol{z}) - \pi_2 f_2(\boldsymbol{z})\, dz$$

$$= \int_{Q(\boldsymbol{z})>0} \pi_1 f_1(\boldsymbol{z}) - \pi_2 f_2(\boldsymbol{z})\, dz$$

$$+ \int_{\widetilde{Q}(\boldsymbol{z})\leq 0} \pi_1 f_1(\boldsymbol{z}) - \pi_2 f_2(\boldsymbol{z})\, dz - (\pi_1 - \pi_2).$$

Therefore,

$$R(G_{\widetilde{Q}}) - R(G_Q) = \int_{Q(\boldsymbol{z})>0, \widetilde{Q}(\boldsymbol{z})\leq 0} \pi_1 f_1(\boldsymbol{z}) - \pi_2 f_2(\boldsymbol{z})\, dz$$

$$= \int_{Q(\boldsymbol{z})>0, \widetilde{Q}(\boldsymbol{z})\leq 0} \pi_1 f_1(\boldsymbol{z}) \left( -\frac{\pi_1 f_1(\boldsymbol{z})}{\pi_2 f_2(\boldsymbol{z})} + 1 \right) dz$$

$$= \int_{Q(\boldsymbol{z})>0, Q(\boldsymbol{z})\leq Q(\boldsymbol{z})-\widetilde{Q}(\boldsymbol{z})} \pi_1 f_1(\boldsymbol{z}) \left( -e^{(\log(f_1(\boldsymbol{z})) - \log(f_2(\boldsymbol{z})) + \log(\frac{\pi_1}{\pi_2}))} + 1 \right) dz.$$

Let

$$\log(f_1(\boldsymbol{z})) - \log(f_2(\boldsymbol{z}))$$

$$= \log \left( \frac{g((\boldsymbol{z} - \boldsymbol{\mu}_2)^T \boldsymbol{\Sigma}_1^{-1}(\boldsymbol{z} - \boldsymbol{\mu}_2))}{g((\boldsymbol{z} - \boldsymbol{\mu}_2)^T \boldsymbol{\Sigma}_2^{-1}(\boldsymbol{z} - \boldsymbol{\mu}_2))} \right) - \frac{1}{2} \log \frac{|\boldsymbol{\Sigma}_1|}{|\boldsymbol{\Sigma}_2|}$$

$$:= \frac{1}{2} Q_E(\boldsymbol{z}).$$

Then,

$$
\begin{aligned}
& R(G_{\widetilde{Q}}) - R(G_Q) \\
&= \int_{Q(\boldsymbol{z})>0,Q(\boldsymbol{z})\leq Q(\boldsymbol{z})-\widetilde{Q}(\boldsymbol{z})} \pi_1 f_1(\boldsymbol{z}) \left( -e^{\frac{Q_E(\boldsymbol{z})}{2}} + 1 \right) dz \\
&= \frac{1}{2} \mathbb{E}_{\boldsymbol{z}\sim f_1}[(1 - e^{\frac{Q_E(\boldsymbol{z})}{2}})\mathbb{1}\{Q(\boldsymbol{z}) > 0, Q(\boldsymbol{z}) \leq Q(\boldsymbol{z}) - \widetilde{Q}(\boldsymbol{z})\}].
\end{aligned} \tag{12}
$$

Let $M(\boldsymbol{z}) = Q(\boldsymbol{z}) - \widetilde{Q}(\boldsymbol{z})$ . We next consider the tail probability of $M(\boldsymbol{z})$ when $\boldsymbol{z} \sim f_1$ .We can first rewrite the QDA rule in (2) as follows :

$$
\begin{aligned}
Q(\boldsymbol{z}) &= (\boldsymbol{z}-\boldsymbol{\mu}_1)^T D(\boldsymbol{z}-\boldsymbol{\mu}_1) - 2\boldsymbol{\beta}^T(\boldsymbol{z}-\bar{\boldsymbol{\mu}}) - \log(|\mathbf{D}\boldsymbol{\Sigma}_1 + \mathbf{I}_p|) \\
&= (\boldsymbol{z}-\boldsymbol{\mu}_1)^T D(\boldsymbol{z}-\boldsymbol{\mu}_1) - 2\boldsymbol{\beta}^T(\boldsymbol{z}-\boldsymbol{\mu}_1) + \boldsymbol{\beta}^T(\boldsymbol{\mu}_2-\boldsymbol{\mu}_1) - \log(|\mathbf{D}\boldsymbol{\Sigma}_1 + \mathbf{I}_p|).
\end{aligned}
$$

Consider the const term first. With probability at least $1 - O\left(\frac{1}{\log p}\right)$, we have

$$
\left| \boldsymbol{\beta}^\top(\boldsymbol{\mu}_2 - \boldsymbol{\mu}_1) - \tilde{\boldsymbol{\beta}}^\top(\tilde{\boldsymbol{\mu}}_2 - \tilde{\boldsymbol{\mu}}_1) \right|
$$

$$
\leq \left| \tilde{\boldsymbol{\beta}}^\top(\boldsymbol{\mu}_2 - \boldsymbol{\mu}_1 - \tilde{\boldsymbol{\mu}}_2 + \tilde{\boldsymbol{\mu}}_1) \right| + \left\| (\tilde{\boldsymbol{\beta}} - \boldsymbol{\beta})^\top(\boldsymbol{\mu}_2 - \boldsymbol{\mu}_1) \right\|_2
$$

$$
\leq \|\tilde{\boldsymbol{\beta}}\|_1 \cdot \|\boldsymbol{\mu}_2 - \boldsymbol{\mu}_1 - \tilde{\boldsymbol{\mu}}_2 + \tilde{\boldsymbol{\mu}}_1\|_\infty + \|\tilde{\boldsymbol{\beta}} - \boldsymbol{\beta}\|_2\|\boldsymbol{\mu}_2 - \boldsymbol{\mu}_1\|_2
$$

$$
\leq \|\boldsymbol{\beta}\|_1 \cdot \|\boldsymbol{\mu}_2 - \boldsymbol{\mu}_1 - \tilde{\boldsymbol{\mu}}_2 + \tilde{\boldsymbol{\mu}}_1\|_\infty + \|\tilde{\boldsymbol{\beta}} - \boldsymbol{\beta}\|_2\|\boldsymbol{\mu}_2 - \boldsymbol{\mu}_1\|_2
$$

$$
\leq \sqrt{s_2}\|\boldsymbol{\beta}\|_2 \cdot \|\boldsymbol{\mu}_2 - \boldsymbol{\mu}_1 - \tilde{\boldsymbol{\mu}}_2 + \tilde{\boldsymbol{\mu}}_1\|_\infty + \|\tilde{\boldsymbol{\beta}} - \boldsymbol{\beta}\|_2\|\boldsymbol{\mu}_2 - \boldsymbol{\mu}_1\|_2 \lesssim s_2 \left( \sqrt{\frac{1}{p}} + \sqrt{\frac{\log(p)}{n}} \right).
$$

Next, we consider $\log |\tilde{\mathbf{D}}\tilde{\boldsymbol{\Sigma}}_1 + \mathbf{I}_p| - \log |\mathbf{D}\boldsymbol{\Sigma}_1 + \mathbf{I}_p|$. Since $(\mathbf{D}\boldsymbol{\Sigma}_1 + \mathbf{I}_p)^{-1} = \boldsymbol{\Omega}_1\boldsymbol{\Sigma}_2 = (\boldsymbol{\Omega}_2 - \mathbf{D})\boldsymbol{\Sigma}_2 = \mathbf{I}_p - \mathbf{D}\boldsymbol{\Sigma}_2$. By Lemma A.9 ,

$$
\begin{aligned}
& \log |\tilde{\mathbf{D}}\tilde{\boldsymbol{\Sigma}}_1 + \mathbf{I}_p| - \log |\mathbf{D}\boldsymbol{\Sigma}_1 + \mathbf{I}_p| \\
&\leq \mathrm{tr}((\mathbf{D}\boldsymbol{\Sigma}_1 + \mathbf{I}_p)^{-1}(\tilde{\mathbf{D}}\tilde{\boldsymbol{\Sigma}}_1 - \mathbf{D}\boldsymbol{\Sigma}_1)) \\
&= \mathrm{tr}((-\mathbf{D}\boldsymbol{\Sigma}_2 + \mathbf{I}_p)(\tilde{\mathbf{D}}\tilde{\boldsymbol{\Sigma}}_1 - \mathbf{D}\boldsymbol{\Sigma}_1)) \\
&= \mathrm{tr}((-\mathbf{D}\boldsymbol{\Sigma}_2)(\tilde{\mathbf{D}}\tilde{\boldsymbol{\Sigma}}_1 - \mathbf{D}\boldsymbol{\Sigma}_1)) + \mathrm{tr}(\tilde{\mathbf{D}}\tilde{\boldsymbol{\Sigma}}_1 - \mathbf{D}\boldsymbol{\Sigma}_1) \\
&\leq \|\mathbf{D}\boldsymbol{\Sigma}_2\|_F \cdot \|\tilde{\mathbf{D}}\tilde{\boldsymbol{\Sigma}}_1 - \mathbf{D}\boldsymbol{\Sigma}_1\|_F + \mathrm{tr}(\tilde{\mathbf{D}}\tilde{\boldsymbol{\Sigma}}_1 - \mathbf{D}\boldsymbol{\Sigma}_1) \\
&\leq \|\mathbf{D}\|_F\|\boldsymbol{\Sigma}_2\|_2 \cdot \|\tilde{\mathbf{D}}\tilde{\boldsymbol{\Sigma}}_1 - \mathbf{D}\boldsymbol{\Sigma}_1\|_F + \mathrm{tr}(\tilde{\mathbf{D}}\tilde{\boldsymbol{\Sigma}}_1 - \mathbf{D}\boldsymbol{\Sigma}_1) \\
&\leq \|\mathbf{D}\|_F\|\boldsymbol{\Sigma}_2\|_2 \cdot \|\tilde{\mathbf{D}}\tilde{\boldsymbol{\Sigma}}_1 - \mathbf{D}\boldsymbol{\Sigma}_1\|_F + |\mathrm{tr}(\tilde{\mathbf{D}}\tilde{\boldsymbol{\Sigma}}_1 - \tilde{\mathbf{D}}\boldsymbol{\Sigma}_1)| + \mathrm{tr}(\tilde{\mathbf{D}}\boldsymbol{\Sigma}_1 - \mathbf{D}\boldsymbol{\Sigma}_1),
\end{aligned}
$$

where

$$
\begin{aligned}
& \left\| \mathbf{D}\boldsymbol{\Sigma}_1 - \tilde{\mathbf{D}}\tilde{\boldsymbol{\Sigma}}_1 \right\|_F \\
&\leq \left\| \mathbf{D}\boldsymbol{\Sigma}_1 - \tilde{\mathbf{D}}\boldsymbol{\Sigma}_1 \right\|_F + \left\| \tilde{\mathbf{D}}(\boldsymbol{\Sigma}_1 - \tilde{\boldsymbol{\Sigma}}_1) \right\|_F \\
&\leq \|\mathbf{D} - \tilde{\mathbf{D}}\|_F\|\boldsymbol{\Sigma}_1\|_2 + \|\tilde{\mathbf{D}}\|_F\|\boldsymbol{\Sigma}_1 - \tilde{\boldsymbol{\Sigma}}_1\|_{2,s_1}.
\end{aligned}
$$

Since

$$
\begin{aligned}
\|\boldsymbol{\Sigma}_1 - \tilde{\boldsymbol{\Sigma}}_1\|_{2,s_1} &= \sup_{\|\boldsymbol{u}\|_0 \leq s_1, \|\boldsymbol{u}\|_2 = 1} \|(\boldsymbol{\Sigma}_1 - \tilde{\boldsymbol{\Sigma}}_1)\boldsymbol{u}\|_2 \\
&= \sup_{\|\boldsymbol{u}\|_0 \leq s_1, \|\boldsymbol{u}\|_2 = 1} |\boldsymbol{u}^T(\boldsymbol{\Sigma}_1 - \tilde{\boldsymbol{\Sigma}}_1)\boldsymbol{u}|,
\end{aligned}
$$

by triangle inequality, we can obtain

$$
\begin{aligned}
\|\boldsymbol{\Sigma}_1 - \tilde{\boldsymbol{\Sigma}}_1\|_{2,s_1} &\leq \frac{\mathrm{tr}(\boldsymbol{\Sigma}_1)}{p} \left\{ \left\| \left( \frac{\widetilde{\mathrm{tr}(\boldsymbol{\Sigma}_1)}}{\mathrm{tr}(\boldsymbol{\Sigma}_1)} - 1 \right) p\tilde{\mathbf{S}}_1 \right\|_{2,s_1} + \|p\tilde{\mathbf{S}}_1 - p\mathbf{S}_1\|_{2,s_1} + \|p\mathbf{S}_1 - \boldsymbol{\Lambda}_1\|_{2,s_1} \right\} \\
&\lesssim \left| \frac{\widetilde{\mathrm{tr}(\boldsymbol{\Sigma}_1)}}{\mathrm{tr}(\boldsymbol{\Sigma}_1)} - 1 \right| \left\| p\tilde{\mathbf{S}}_1 \right\|_{2,s_1} + \|p\tilde{\mathbf{S}}_1 - p\mathbf{S}_1\|_{2,s_1} + \|p\mathbf{S}_1 - \boldsymbol{\Lambda}_1\|_{2,s_1}.
\end{aligned}
$$

With (10) and Lemma A.7, $\|p\tilde{\mathbf{S}}_1 - p\mathbf{S}_1\|_{2,s_1} \lesssim \sqrt{\frac{s_1 \log p}{n}}$, $\|pS\|_{2,s_1} \le \|pS\|_2 \lesssim 1$ with probability over $1 - 3/p$. Therefore, by Lemma A.4, we obtain

$$P\left( \left| \frac{\widetilde{\text{tr}(\boldsymbol{\Sigma}_1)}}{\text{tr}(\boldsymbol{\Sigma}_1)} - 1 \right| \left\| p\tilde{\mathbf{S}}_1 \right\|_{2,s_1} \ge \sqrt{\frac{\log p}{n}} \right)$$

$$\le P\left( \left| \frac{\widetilde{\text{tr}(\boldsymbol{\Sigma}_1)}}{\text{tr}(\boldsymbol{\Sigma}_1)} - 1 \right| \left\| p\tilde{\mathbf{S}}_1 \right\|_{2,s_1} \ge \sqrt{\frac{\log p}{n}} \left| \|p\mathbf{S}_1\|_{2,s_1} \le C \right. \right) + P\left( \|p\mathbf{S}_1\|_{2,s_1} \ge C \right)$$

$$\lesssim \frac{1}{\log p} + \frac{1}{p} \lesssim \frac{1}{\log p}.$$

According to Lemma A.11 , $\|p\mathbf{S}_1 - \boldsymbol{\Lambda}_1\|_{2,s_1} \le s_1 \|p\mathbf{S}_1 - \boldsymbol{\Lambda}_1\|_{\max} \lesssim \frac{s_1}{\sqrt{p}}$ . Therefore, with probability over $1 - O\left( \frac{1}{\log p} \right)$,

$$\|\mathbf{D}\boldsymbol{\Sigma}_1 - \tilde{\mathbf{D}}\tilde{\boldsymbol{\Sigma}}_1\|_F \lesssim s_1 \left( \sqrt{\frac{1}{p}} + \sqrt{\frac{\log(p)}{n}} \right).$$

For the second term $\left| \text{tr}\left( \tilde{\mathbf{D}}\boldsymbol{\Sigma}_1 - \tilde{\mathbf{D}}\tilde{\boldsymbol{\Sigma}}_1 \right) \right|$ , with probability at least $1 - O\left( \frac{1}{\log p} \right)$,

$$\left| \text{tr}\left( \tilde{\mathbf{D}}\boldsymbol{\Sigma}_1 - \tilde{\mathbf{D}}\tilde{\boldsymbol{\Sigma}}_1 \right) \right| \le \|\boldsymbol{\Sigma}_1 - \tilde{\boldsymbol{\Sigma}}_1\|_{max} \sqrt{s_1} \|\tilde{\mathbf{D}}\|_F \le s_1 \left( \sqrt{\frac{1}{p}} + \sqrt{\frac{\log(p)}{n}} \right).$$

Therefore , with probability over $1 - O\left( \frac{1}{\log p} \right)$,

$$\log|\tilde{\mathbf{D}}\tilde{\boldsymbol{\Sigma}}_1 + \mathbf{I}_p| - \log|\mathbf{D}\boldsymbol{\Sigma}_1 + \mathbf{I}_p| - \text{tr}(\tilde{\mathbf{D}}\tilde{\boldsymbol{\Sigma}}_1 - \mathbf{D}\boldsymbol{\Sigma}_1)$$

$$\lesssim s_1 \left( \sqrt{\frac{1}{p}} + \sqrt{\frac{\log p}{n}} \right).$$

As for the other side, by Lemma A.9,

$$\log|\mathbf{D}\boldsymbol{\Sigma}_1 + I_P| - \log|\tilde{\mathbf{D}}\tilde{\boldsymbol{\Sigma}}_1 + \mathbf{I}_p|$$

$$\le \text{tr}((\tilde{\mathbf{D}}\tilde{\boldsymbol{\Sigma}}_1 + \mathbf{I}_p)^{-1}(\mathbf{D}\boldsymbol{\Sigma}_1 - \tilde{\mathbf{D}}\tilde{\boldsymbol{\Sigma}}_1))$$

$$\le \text{tr}([(\tilde{\mathbf{D}}\tilde{\boldsymbol{\Sigma}}_1 + \mathbf{I}_p)^{-1} - (\mathbf{D}\boldsymbol{\Sigma}_1 + \mathbf{I}_p)^{-1}](\mathbf{D}\boldsymbol{\Sigma}_1 - \tilde{\mathbf{D}}\tilde{\boldsymbol{\Sigma}}_1)) + \text{tr}((\mathbf{D}\boldsymbol{\Sigma}_1 + \mathbf{I}_p)^{-1}(\mathbf{D}\boldsymbol{\Sigma}_1 - \tilde{\mathbf{D}}\tilde{\boldsymbol{\Sigma}}_1))$$

$$\le \text{tr}([(\tilde{\mathbf{D}}\tilde{\boldsymbol{\Sigma}}_1 + \mathbf{I}_p)^{-1} - (\mathbf{D}\boldsymbol{\Sigma}_1 + \mathbf{I}_p)^{-1}](\mathbf{D}\boldsymbol{\Sigma}_1 - \tilde{\mathbf{D}}\tilde{\boldsymbol{\Sigma}}_1)) + \|\mathbf{D}\boldsymbol{\Sigma}_2\|_F \cdot \|\tilde{\mathbf{D}}\tilde{\boldsymbol{\Sigma}}_1 - \mathbf{D}\boldsymbol{\Sigma}_1\|_F$$

$$\quad + \text{tr}(\mathbf{D}\boldsymbol{\Sigma}_1 - \tilde{\mathbf{D}}\tilde{\boldsymbol{\Sigma}}_1)$$

$$\le \text{tr}([(\tilde{\mathbf{D}}\tilde{\boldsymbol{\Sigma}}_1 + \mathbf{I}_p)^{-1} - (\mathbf{D}\boldsymbol{\Sigma}_1 + \mathbf{I}_p)^{-1}](\mathbf{D}\boldsymbol{\Sigma}_1 - \tilde{\mathbf{D}}\tilde{\boldsymbol{\Sigma}}_1)) +$$

$$\quad \|\mathbf{D}\boldsymbol{\Sigma}_2\|_F \cdot \|\tilde{\mathbf{D}}\tilde{\boldsymbol{\Sigma}}_1 - \mathbf{D}\boldsymbol{\Sigma}_1\|_F + |\text{tr}(\tilde{\mathbf{D}}\boldsymbol{\Sigma}_1 - \tilde{\mathbf{D}}\tilde{\boldsymbol{\Sigma}}_1)| - \text{tr}(\tilde{\mathbf{D}}\boldsymbol{\Sigma}_1 - \mathbf{D}\boldsymbol{\Sigma}_1)$$

$$\lesssim \text{tr}([(\tilde{\mathbf{D}}\tilde{\boldsymbol{\Sigma}}_1 + \mathbf{I}_p)^{-1} - (\mathbf{D}\boldsymbol{\Sigma}_1 + \mathbf{I}_p)^{-1}](\mathbf{D}\boldsymbol{\Sigma}_1 - \tilde{\mathbf{D}}\tilde{\boldsymbol{\Sigma}}_1))$$

$$\quad + s_1 \left( \sqrt{\frac{1}{p}} + \sqrt{\frac{\log p}{n}} \right) - \text{tr}(\tilde{\mathbf{D}}\boldsymbol{\Sigma}_1 - \mathbf{D}\boldsymbol{\Sigma}_1).$$

Consider

$$\text{tr}([(\tilde{\mathbf{D}}\tilde{\boldsymbol{\Sigma}}_1 + \mathbf{I}_p)^{-1} - (\mathbf{D}\boldsymbol{\Sigma}_1 + \mathbf{I}_p)^{-1}](\mathbf{D}\boldsymbol{\Sigma}_1 - \tilde{\mathbf{D}}\tilde{\boldsymbol{\Sigma}}_1))$$

$$\le \|(\tilde{\mathbf{D}}\tilde{\boldsymbol{\Sigma}}_1 + \mathbf{I}_p)^{-1} - (\mathbf{D}\boldsymbol{\Sigma}_1 + \mathbf{I}_p)^{-1}\|_F \cdot \|(\mathbf{D}\boldsymbol{\Sigma}_1 - \tilde{\mathbf{D}}\tilde{\boldsymbol{\Sigma}}_1)\|_F.$$

Let $A := \tilde{\mathbf{D}}\tilde{\boldsymbol{\Sigma}}_1 + \mathbf{I}_p, B := \mathbf{D}\boldsymbol{\Sigma}_1 + \mathbf{I}_p$ , then

$$\|\mathbf{A}^{-1} - \mathbf{B}^{-1}\|_F = \|\mathbf{A}^{-1}(\mathbf{B} - \mathbf{A})\mathbf{B}^{-1}\|_F$$

$$\le \|\mathbf{A}^{-1}\|_2 \|(\mathbf{B} - \mathbf{A})\mathbf{B}^{-1}\|_F$$

$$\le \|\mathbf{A}^{-1}\|_2 \cdot \|(\mathbf{B} - \mathbf{A})\|_F \cdot \|\mathbf{B}^{-1}\|_2,$$

where
$$\|\mathbf{B}^{-1}\|_2 = \|\mathbf{I}_p - \mathbf{D}\boldsymbol{\Sigma}_2\|_2 \leq 1 + \|\mathbf{D}\boldsymbol{\Sigma}_2\|_2$$
$$\leq 1 + \|\mathbf{D}\|_F \|\boldsymbol{\Sigma}_2\|_2 := M.$$

From Lemma A.10
$$\|\mathbf{A}^{-1}\|_2 = \|[(\mathbf{I} + (\mathbf{A} - \mathbf{B})\mathbf{B}^{-1})\mathbf{B}]^{-1}\|_2$$
$$= \|\mathbf{B}^{-1}(\mathbf{I} + (\mathbf{A} - \mathbf{B})\mathbf{B}^{-1})^{-1}\|_2$$
$$\leq \|\mathbf{B}^{-1}\|_2 \|(\mathbf{I} - (\mathbf{B} - \mathbf{A})\mathbf{B}^{-1})^{-1}\|_2.$$

Let $\mathbf{E} := (\mathbf{B} - \mathbf{A})\mathbf{B}^{-1}$ , when $n, p$ is large enough
$$\|\mathbf{E}\|_2 \leq \|\mathbf{B} - \mathbf{A}\|_2 \|\mathbf{B}^{-1}\|_2 \leq \|\mathbf{B} - \mathbf{A}\|_F M$$
$$\lesssim s_1 \left( \sqrt{\frac{1}{p}} + \sqrt{\frac{\log p}{n}} \right) \quad < 1.$$

With Lemma A.10 , when $n$ is sufficient large ,
$$\|(\mathbf{I} - \mathbf{E})^{-1}\|_2 \leq \frac{1}{1 - \|\mathbf{E}\|_2} < M + 1.$$

Thus $\|\mathbf{A}^{-1}\|_2$ is bounded, and $\|\mathbf{A}^{-1} - \mathbf{B}^{-1}\|_F \lesssim \|\mathbf{B} - \mathbf{A}\|_F$ . Therefore,
$$\mathrm{tr}([(\tilde{\mathbf{D}}\tilde{\boldsymbol{\Sigma}}_1 + \mathbf{I}_p)^{-1} - (\mathbf{D}\boldsymbol{\Sigma}_1 + \mathbf{I}_p)^{-1}](\mathbf{D}\boldsymbol{\Sigma}_1 - \tilde{\mathbf{D}}\tilde{\boldsymbol{\Sigma}}_1))$$
$$\lesssim \|(\mathbf{D}\boldsymbol{\Sigma}_1 - \tilde{\mathbf{D}}\tilde{\boldsymbol{\Sigma}}_1)\|_F$$
$$\lesssim s_1 \left( \sqrt{\frac{1}{p}} + \sqrt{\frac{\log p}{n}} \right).$$

Consequently, we have that with probability $1 - O\left(\frac{1}{\log p}\right)$,
$$|\log|\tilde{\mathbf{D}}\tilde{\boldsymbol{\Sigma}}_1 + \mathbf{I}_p| - \log|\mathbf{D}\boldsymbol{\Sigma}_1 + \mathbf{I}_p| - \mathrm{tr}(\tilde{\mathbf{D}}\tilde{\boldsymbol{\Sigma}}_1 - \mathbf{D}\boldsymbol{\Sigma}_1)|$$
$$\lesssim s_1 \left( \sqrt{\frac{1}{p}} + \sqrt{\frac{\log p}{n}} \right).$$

Next we consider the term involving $\boldsymbol{z}$. That is $(\boldsymbol{z} - \boldsymbol{\mu}_1)^T \mathbf{D}(\boldsymbol{z} - \boldsymbol{\mu}_1) - (\boldsymbol{z} - \boldsymbol{\mu}_1)^T \tilde{\mathbf{D}}(\boldsymbol{z} - \boldsymbol{\mu}_1) - (\mathrm{tr}(\mathbf{D}\boldsymbol{\Sigma}_1 - \tilde{\mathbf{D}}\tilde{\boldsymbol{\Sigma}}_1))$. Recall the definition of elliptical distribution, for $\boldsymbol{z}$ in class 1, we can assume $\boldsymbol{z} = \boldsymbol{\mu}_1 + r\boldsymbol{\Gamma}_1\boldsymbol{u}$, where $\boldsymbol{u}$ is uniformly distributed on the sphere and $r$ is a scalar random variable with $E(r^2) = p$ and independent with $\boldsymbol{u}$. $\boldsymbol{u}$ could be expressed as $\boldsymbol{u} = \boldsymbol{x}/\|\boldsymbol{x}\|$, where $\boldsymbol{x} \sim N(\mathbf{0}, \mathbf{I}_p)$. So $\boldsymbol{z} = \boldsymbol{\mu}_1 + r\|\boldsymbol{x}\|^{-1}\boldsymbol{\Gamma}_1\boldsymbol{x}$ and
$$(\boldsymbol{z} - \boldsymbol{\mu}_1)^T \mathbf{D}(\boldsymbol{z} - \boldsymbol{\mu}_1) - (\boldsymbol{z} - \boldsymbol{\mu}_1)^T \tilde{\mathbf{D}}(\boldsymbol{z} - \boldsymbol{\mu}_1)$$
$$= r^2 \|\boldsymbol{x}\|^{-2} \boldsymbol{x}^T \boldsymbol{\Gamma}_1^T (\mathbf{D} - \tilde{\mathbf{D}}) \boldsymbol{\Gamma}_1 \boldsymbol{x}$$
$$= [r^2 \|\boldsymbol{x}\|^{-2} - E(r^2 \|\boldsymbol{x}\|^{-2})] \boldsymbol{x}^T \boldsymbol{\Gamma}_1^T (\mathbf{D} - \tilde{\mathbf{D}}) \boldsymbol{\Gamma}_1 \boldsymbol{x} + E(r^2 \|\boldsymbol{x}\|^{-2}) \boldsymbol{x}^T \boldsymbol{\Gamma}_1^T (\mathbf{D} - \tilde{\mathbf{D}}) \boldsymbol{\Gamma}_1 \boldsymbol{x}.$$

Given $\|\boldsymbol{x}\|^2 \sim \chi_p^2$ , we have $E(r^2 \|\boldsymbol{x}\|^{-2}) = p/(p-2)$ and
$$\mathrm{Var}(r^2 \|\boldsymbol{x}\|^{-2}) = Er^4 E(\|\boldsymbol{x}\|^{-4}) - [E(r^2)E(\|\boldsymbol{x}\|^{-2})]^2$$
$$= \frac{E(r^4)}{(p-2)(p-4)} - \frac{p^2}{(p-2)^2}.$$

By Chebyshev's inequality
$$P\left(|r^2\|\boldsymbol{x}\|^{-2} - E(r^2\|\boldsymbol{x}\|^{-2})| > t\right)$$
$$\leq \frac{\mathrm{Var}(r^2\|\boldsymbol{x}\|^{-2})}{t^2}$$
$$= \frac{1}{t^2} \left( \frac{E(r^4)}{(p-2)(p-4)} - \frac{p^2}{(p-2)^2} \right)$$
$$= \frac{1}{t^2} \left( \frac{(p-2)\mathrm{Var}(r^2) + 2p^2}{(p-2)^2(p-4)} \right).$$

As the same in Cai & Zhang (2021),

$$\boldsymbol{x}^T\boldsymbol{\Gamma_1}^T(\mathbf{D}-\tilde{\mathbf{D}})\boldsymbol{\Gamma}_1\boldsymbol{x} - \text{tr}(\boldsymbol{\Gamma_1}^T(\mathbf{D}-\tilde{\mathbf{D}})\boldsymbol{\Gamma}_1) = \sum_{i=1}^{p}\lambda_i(x_i^2-1),$$

where $\lambda_i$ are the eigenvalue of $\boldsymbol{\Gamma_1}^T(\mathbf{D}-\tilde{\mathbf{D}})\boldsymbol{\Gamma}_1$. Since with probability at least $1 - O\left(\frac{1}{\log p}\right)$,

$$\sqrt{\sum_{i=1}^{p}\lambda_i^2} = \|\boldsymbol{\Sigma_1}^{1/2}(\tilde{\mathbf{D}}-\mathbf{D})\boldsymbol{\Sigma_1}^{1/2}\|_F \le \|\boldsymbol{\Sigma}_1\|_2\|\tilde{\mathbf{D}}-\mathbf{D}\|_F \lesssim s_1\left(\sqrt{\frac{1}{p}}+\sqrt{\frac{\log p}{n}}\right),$$

and with probability at least $1 - O\left(\frac{1}{\log p}\right)$,

$$\max_i |\lambda_i| \le \|\boldsymbol{\Sigma_1}^{1/2}(\tilde{\mathbf{D}}-\mathbf{D})\boldsymbol{\Sigma_1}^{1/2}\|_2 \le \|\boldsymbol{\Sigma}_1\|_2\|\tilde{\mathbf{D}}-\mathbf{D}\|_2 \lesssim s_1\left(\sqrt{\frac{1}{p}}+\sqrt{\frac{\log p}{n}}\right).$$

By Bernstein inequality for sub-exponential random variables, we have for some $c_1 > 0$,

$$\mathbb{P}\left(\left|\sum_{i=1}^{p}\lambda_i(x_i^2-1)\right| \ge t\right) \le 2\exp\left\{-c_1\min\left\{\frac{t^2}{s_1^2\left(\sqrt{\frac{1}{p}}+\sqrt{\frac{\log p}{n}}\right)^2}, \frac{t}{s_1\left(\sqrt{\frac{1}{p}}+\sqrt{\frac{\log p}{n}}\right)}\right\}\right\}$$
$$+\frac{C}{\log p}. \tag{13}$$

Thus, we can obtain

$$(\boldsymbol{z}-\boldsymbol{\mu}_1)^T\mathbf{D}(\boldsymbol{z}-\boldsymbol{\mu}_1) - (\boldsymbol{z}-\boldsymbol{\mu}_1)^T\tilde{\mathbf{D}}(\boldsymbol{z}-\boldsymbol{\mu}_1) - (\text{tr}(\mathbf{D}\boldsymbol{\Sigma}_1-\tilde{\mathbf{D}}\boldsymbol{\Sigma}_1))$$
$$=[r^2\|\boldsymbol{x}\|^{-2}-E(r^2\|\boldsymbol{x}\|^{-2})]\boldsymbol{x}^T\boldsymbol{\Gamma_1}^T(\mathbf{D}-\tilde{\mathbf{D}})\boldsymbol{\Gamma}_1\boldsymbol{x}+E(r^2\|\boldsymbol{x}\|^{-2})\boldsymbol{x}^T\boldsymbol{\Gamma_1}^T(\mathbf{D}-\tilde{\mathbf{D}})\boldsymbol{\Gamma}_1\boldsymbol{x}$$
$$- \text{tr}(\mathbf{D}\boldsymbol{\Sigma}_1-\tilde{\mathbf{D}}\boldsymbol{\Sigma}_1)$$
$$=[r^2\|\boldsymbol{x}\|^{-2}-E(r^2\|\boldsymbol{x}\|^{-2})]\left(\boldsymbol{x}^T\boldsymbol{\Gamma_1}^T(\mathbf{D}-\tilde{\mathbf{D}})\boldsymbol{\Gamma}_1-\text{tr}(\boldsymbol{\Gamma_1}^T(\mathbf{D}-\tilde{\mathbf{D}})\boldsymbol{\Gamma}_1)\right)$$
$$+\frac{p}{p-2}\left(\boldsymbol{x}^T\boldsymbol{\Gamma_1}^T(\mathbf{D}-\tilde{\mathbf{D}})\boldsymbol{\Gamma}_1\boldsymbol{x}-\text{tr}(\boldsymbol{\Gamma_1}^T(\mathbf{D}-\tilde{\mathbf{D}})\boldsymbol{\Gamma}_1)\right)$$
$$+\left(\frac{p}{p-2}+r^2\|\boldsymbol{x}\|^{-2}-E(r^2\|\boldsymbol{x}\|^{-2})-1\right)\text{tr}(\boldsymbol{\Gamma_1}^T(\mathbf{D}-\tilde{\mathbf{D}})\boldsymbol{\Gamma}_1).$$

Then, we can consider

$$P\left(\left|(\boldsymbol{z}-\boldsymbol{\mu}_1)^T\mathbf{D}(\boldsymbol{z}-\boldsymbol{\mu}_1) - (\boldsymbol{z}-\boldsymbol{\mu}_1)^T\tilde{\mathbf{D}}(\boldsymbol{z}-\boldsymbol{\mu}_1) - (\text{tr}(\mathbf{D}\boldsymbol{\Sigma}_1-\tilde{\mathbf{D}}\boldsymbol{\Sigma}_1))\right|\right.$$
$$\left.\ge Ms_1\left(\sqrt{\frac{1}{p}}+\sqrt{\frac{\log p}{n}}\right)\right)$$
$$\le P\left(\left|[r^2\|\boldsymbol{x}\|^{-2}-E(r^2\|\boldsymbol{x}\|^{-2})]\left(\boldsymbol{x}^T\boldsymbol{\Gamma_1}^T(\mathbf{D}-\tilde{\mathbf{D}})\boldsymbol{\Gamma}_1-\text{tr}(\boldsymbol{\Gamma_1}^T(\mathbf{D}-\tilde{\mathbf{D}})\boldsymbol{\Gamma}_1)\right)\right|\right.$$
$$\left.\ge \frac{M}{3}s_1\left(\sqrt{\frac{1}{p}}+\sqrt{\frac{\log p}{n}}\right)\right)$$
$$+P\left(\frac{p}{p-2}\left|\boldsymbol{x}^T\boldsymbol{\Gamma_1}^T(\mathbf{D}-\tilde{\mathbf{D}})\boldsymbol{\Gamma}_1\boldsymbol{x}-\text{tr}(\boldsymbol{\Gamma_1}^T(\mathbf{D}-\tilde{\mathbf{D}})\boldsymbol{\Gamma}_1)\right| \ge \frac{M}{3}s_1\left(\sqrt{\frac{1}{p}}+\sqrt{\frac{\log p}{n}}\right)\right)$$
$$+P\left(\left|\left(\frac{p}{p-2}+r^2\|\boldsymbol{x}\|^{-2}-E(r^2\|\boldsymbol{x}\|^{-2})-1\right)\text{tr}(\boldsymbol{\Gamma_1}^T(\mathbf{D}-\tilde{\mathbf{D}})\boldsymbol{\Gamma}_1)\right|\right.$$
$$\left.\ge \frac{M}{3}s_1\left(\sqrt{\frac{1}{p}}+\sqrt{\frac{\log p}{n}}\right)\right).$$

For the first part,

$$P\left(\left|[r^2\|\boldsymbol{x}\|^{-2} - E(r^2\|\boldsymbol{x}\|^{-2})]\left(\boldsymbol{x}^T\boldsymbol{\Gamma}_1{}^T(\mathbf{D} - \tilde{\mathbf{D}})\boldsymbol{\Gamma}_1 - \mathrm{tr}(\boldsymbol{\Gamma}_1{}^T(\mathbf{D} - \tilde{\mathbf{D}})\boldsymbol{\Gamma}_1))\right|\right.$$

$$\left.\geq \frac{M}{3}s_1\left(\sqrt{\frac{1}{p}} + \sqrt{\frac{\log p}{n}}\right)\right)$$

$$\leq P\left(\left|\boldsymbol{x}^T\boldsymbol{\Gamma}_1{}^T(\mathbf{D} - \tilde{\mathbf{D}})\boldsymbol{\Gamma}_1 - \mathrm{tr}(\boldsymbol{\Gamma}_1{}^T(\mathbf{D} - \tilde{\mathbf{D}})\boldsymbol{\Gamma}_1)\right| \geq \frac{M}{3}s_1\left(\sqrt{\frac{1}{p}} + \sqrt{\frac{\log p}{n}}\right)\right)$$

$$+ P\left(\left|r^2\|\boldsymbol{x}\|^{-2} - E(r^2\|\boldsymbol{x}\|^{-2})\right| \geq 1\right)$$

$$\leq 2\exp\left\{-c_1\min\left\{\frac{M^2}{9}, \frac{M}{3}\right\}\right\} + \frac{(p-2)\mathrm{Var}(r^2) + 2p^2}{(p-2)^2(p-4)} + \frac{c_2}{\log p}.$$

For the second part, when $p \geq 3$

$$P\left(\frac{p}{p-2}\left|\boldsymbol{x}^T\boldsymbol{\Gamma}_1{}^T(\mathbf{D} - \tilde{\mathbf{D}})\boldsymbol{\Gamma}_1\boldsymbol{x} - \mathrm{tr}(\boldsymbol{\Gamma}_1{}^T(\mathbf{D} - \tilde{\mathbf{D}})\boldsymbol{\Gamma}_1)\right| \geq \frac{M}{3}s_1\left(\sqrt{\frac{1}{p}} + \sqrt{\frac{\log p}{n}}\right)\right)$$

$$\leq P\left(\left|\boldsymbol{x}^T\boldsymbol{\Gamma}_1{}^T(\mathbf{D} - \tilde{\mathbf{D}})\boldsymbol{\Gamma}_1\boldsymbol{x} - \mathrm{tr}(\boldsymbol{\Gamma}_1{}^T(\mathbf{D} - \tilde{\mathbf{D}})\boldsymbol{\Gamma}_1)\right| \geq \frac{M}{9}s_1\left(\sqrt{\frac{1}{p}} + \sqrt{\frac{\log p}{n}}\right)\right)$$

$$\leq 2\exp\left\{-c_1\min\left\{\frac{M^2}{81}, \frac{M}{9}\right\}\right\} + \frac{c_2}{\log p}.$$

For the third part, since with a probability of $1 - O\left(\frac{1}{\log p}\right)$,

$$\left|\mathrm{tr}\left(\boldsymbol{\Gamma}_1{}^T(\mathbf{D} - \tilde{\mathbf{D}})\boldsymbol{\Gamma}_1\right)\right| \leq \|\boldsymbol{\Sigma}_1\|_{\max}\left\|\mathrm{Vec}(\mathbf{D} - \tilde{\mathbf{D}})\right\|_1$$

$$\leq \|\boldsymbol{\Sigma}_1\|_{\max} \cdot 2\sqrt{s_1}\left\|\mathbf{D} - \tilde{\mathbf{D}}\right\|_F$$

$$\lesssim s_1\sqrt{\frac{s_1\log p}{n}}.$$

Therefore,

$$P\left(\left|\left(\frac{2}{p-2} + r^2\|\boldsymbol{x}\|^{-2} - E(r^2\|\boldsymbol{x}\|^{-2}) - 1\right)\sum_{i=1}^p \lambda_i\right| \geq \frac{M}{3}s_1\left(\sqrt{\frac{1}{p}} + \sqrt{\frac{\log p}{n}}\right)\right)$$

$$\leq P\left(\left|\left(\frac{2}{p-2}\right)\sum_{i=1}^p \lambda_i\right| \geq \frac{M}{6}s_1\left(\sqrt{\frac{1}{p}} + \sqrt{\frac{\log p}{n}}\right)\right)$$

$$+ P\left(\left|(r^2\|\boldsymbol{x}\|^{-2} - E(r^2\|\boldsymbol{x}\|^{-2}))\sum_{i=1}^p \lambda_i\right| \geq \frac{M}{6}s_1\left(\sqrt{\frac{1}{p}} + \sqrt{\frac{\log p}{n}}\right)\right)$$

$$\lesssim \frac{1}{p} + P\left(\left|\sum_{i=1}^p \lambda_i\right| \geq \frac{M}{6}s_1\sqrt{\frac{s_1\log p}{n}}\right) + P\left(\left|r^2\|\boldsymbol{x}\|^{-2} - E(r^2\|\boldsymbol{x}\|^{-2})\right| \geq \frac{1}{\sqrt{s_1}}\right)$$

$$\leq \frac{c_2}{\log p} + s_1\frac{(p-2)\mathrm{Var}(r^2) + 2p^2}{(p-2)^2(p-4)}.$$

Thus, there exists constants $c_i$,

$$P\left(\left|(\boldsymbol{z} - \boldsymbol{\mu}_1)^T\mathbf{D}(\boldsymbol{z} - \boldsymbol{\mu}_1) - (\boldsymbol{z} - \boldsymbol{\mu}_1)^T\tilde{\mathbf{D}}(\boldsymbol{z} - \boldsymbol{\mu}_1) - (\mathrm{tr}(\mathbf{D}\boldsymbol{\Sigma}_1 - \tilde{\mathbf{D}}\boldsymbol{\Sigma}_1))\right|\right.$$

$$\left.\geq Ms_1\left(\sqrt{\frac{1}{p}} + \sqrt{\frac{\log p}{n}}\right)\right)$$

$$\leq c_1\exp\{-c_2M\} + \frac{c_3}{\log p} + \frac{(p-2)\mathrm{Var}(r^2) + 2p^2}{(p-2)^2(p-4)} + s_1\frac{(p-2)\mathrm{Var}(r^2) + 2p^2}{(p-2)^2(p-4)}.$$

By Assumption 3.7,

$$P\left(\left|(\boldsymbol{z} - \boldsymbol{\mu}_1)^T \mathbf{D}(\boldsymbol{z} - \boldsymbol{\mu}_1) - (\boldsymbol{z} - \boldsymbol{\mu}_1)^T \tilde{\mathbf{D}}(\boldsymbol{z} - \boldsymbol{\mu}_1) - (\operatorname{tr}(\mathbf{D}\boldsymbol{\Sigma}_1 - \tilde{\mathbf{D}}\boldsymbol{\Sigma}_1))\right|\right.$$

$$\left. \geq M\sqrt{s_1 \frac{\log p}{n}}\right)$$

$$\lesssim \exp\left\{-c_1 M\right\} + \frac{1}{\log p} + \frac{s_1}{\sqrt{p}}.$$

As for the linear term involving $\boldsymbol{z}$,

$$\left|\left(\tilde{\boldsymbol{\beta}} - \boldsymbol{\beta}\right)^T (\boldsymbol{z} - \boldsymbol{\mu}_1)\right| = \left|r\|\boldsymbol{x}\|^{-1}\left(\tilde{\boldsymbol{\beta}} - \boldsymbol{\beta}\right)^T \boldsymbol{\Gamma}_1 \boldsymbol{x}\right|$$

$$\leq \left|[r\|\boldsymbol{x}\|^{-1} - E(r\|\boldsymbol{x}\|^{-1})]\left(\tilde{\boldsymbol{\beta}} - \boldsymbol{\beta}\right)^T \boldsymbol{\Gamma}_1 \boldsymbol{x}\right|$$

$$+ \left|E(r\|\boldsymbol{x}\|^{-1})\left(\tilde{\boldsymbol{\beta}} - \boldsymbol{\beta}\right)^T \boldsymbol{\Gamma}_1 \boldsymbol{x}\right|.$$

Since $\left(\tilde{\boldsymbol{\beta}} - \boldsymbol{\beta}\right)\boldsymbol{\Gamma}_1\boldsymbol{x} \sim N\left(0, \left(\tilde{\boldsymbol{\beta}} - \boldsymbol{\beta}\right)^T\boldsymbol{\Sigma}_1\left(\tilde{\boldsymbol{\beta}} - \boldsymbol{\beta}\right)\right)$, and with probability of $1 - O\left(\frac{1}{\log p}\right)$,

$$\left(\tilde{\boldsymbol{\beta}} - \boldsymbol{\beta}\right)^T\boldsymbol{\Sigma}_1\left(\tilde{\boldsymbol{\beta}} - \boldsymbol{\beta}\right) \leq \|\boldsymbol{\Sigma}_1\|_2\|\tilde{\boldsymbol{\beta}} - \boldsymbol{\beta}\|_2^2 \lesssim s_2^2\frac{\log p}{n}. \tag{14}$$

In addition,

$$P\left(\left|\left(\tilde{\boldsymbol{\beta}} - \boldsymbol{\beta}\right)\boldsymbol{\Gamma}_1\boldsymbol{x}\right| \geq Ms_2\left(\sqrt{\frac{1}{p}} + \sqrt{\frac{\log p}{n}}\right)\Bigg| \tilde{\boldsymbol{\beta}}\right)$$

$$\leq \frac{\sqrt{\left(\tilde{\boldsymbol{\beta}} - \boldsymbol{\beta}\right)^T\boldsymbol{\Sigma}_1\left(\tilde{\boldsymbol{\beta}} - \boldsymbol{\beta}\right)}}{Ms_2\left(\sqrt{\frac{1}{p}} + \sqrt{\frac{\log p}{n}}\right)}\exp\left\{\frac{-s_2\frac{\log p}{n}M^2}{2\left(\tilde{\boldsymbol{\beta}} - \boldsymbol{\beta}\right)^T\boldsymbol{\Sigma}_1\left(\tilde{\boldsymbol{\beta}} - \boldsymbol{\beta}\right)}\right\}.$$

Together with (14),

$$P\left(\left|\left(\tilde{\boldsymbol{\beta}} - \boldsymbol{\beta}\right)\boldsymbol{\Gamma}_1\boldsymbol{x}\right| \geq Ms_2\left(\sqrt{\frac{1}{p}} + \sqrt{\frac{\log p}{n}}\right)\right) \lesssim \exp{-cM^2} + \frac{1}{\log p}.$$

Since $\frac{1}{\sqrt{x}}$ is a convex function, by Jenson's inequality, $E\left(\|\boldsymbol{x}\|^{-1}\right) \geq \sqrt{\frac{1}{E(\|\boldsymbol{x}\|^2)}} = \frac{1}{\sqrt{p}}$. As a result,

$$\operatorname{Var}(r\|\boldsymbol{x}\|^{-1}) = E\left(r^2\|\boldsymbol{x}\|^{-2}\right) - \left(E(r\|\boldsymbol{x}\|^{-1})\right)^2$$

$$= E\left(r^2\|\boldsymbol{x}\|^{-2}\right) - (E(r))^2\left(E(\|\boldsymbol{x}\|^{-1})\right)^2$$

$$\leq \frac{p}{p-2} - \frac{(E(r))^2}{p}$$

$$= \frac{(p-2)\operatorname{Var}(r) + 2p}{p(p-2)}.$$

By Assumption 3.7 and Chebyshev's inequality, we have

$$P\left(\left|r\|\boldsymbol{x}\|^{-1} - E(r\|\boldsymbol{x}\|^{-1})\right| \geq t\right) \leq \frac{(p-2)\operatorname{Var}(r) + 2p}{t^2 p(p-2)}$$

$$\lesssim \frac{1}{t^2\sqrt{p}}.$$

With $E\left(r\|\boldsymbol{x}\|^{-1}\right) \leq \sqrt{E\left(r^2\|\boldsymbol{x}\|^{-2}\right)} = \sqrt{\frac{p}{p-2}}$, there exists constant $c_2$, so that

$$P\left(\left|\left(\tilde{\boldsymbol{\beta}} - \boldsymbol{\beta}\right)^T(\boldsymbol{z} - \boldsymbol{\mu}_1)\right| \geq Ms_2\left(\sqrt{\frac{1}{p}} + \sqrt{\frac{\log p}{n}}\right)\right) \lesssim \exp\{-c_2 M^2\} + \frac{1}{\log p} + \frac{1}{\sqrt{p}}.$$

To complete our analysis, we focus again on the classification error rate. Recall (12) , we have

$$
\begin{aligned}
R(G_{\widetilde{Q}}) - R(G_Q) =& \frac{1}{2}\mathbb{E}_{\boldsymbol{z}\sim f_1}\left[(1 - e^{\frac{Q_E(\boldsymbol{z})}{2}})\mathbb{1}\left\{Q(\boldsymbol{z}) > 0, Q(\boldsymbol{z}) \le Q(\boldsymbol{z}) - \widetilde{Q}(\boldsymbol{z})\right\}\right] \\
\le& \mathbb{E}_{\boldsymbol{z}\sim f_1}\left[\mathbb{1}\left\{Q(\boldsymbol{z}) > 0, Q(\boldsymbol{z}) \le M(\boldsymbol{z})\right\}\right] \\
=& P\left(0 \le Q(\boldsymbol{z}) \le M(\boldsymbol{z})\right) \\
\le& P\left(0 \le Q(\boldsymbol{z}) \le M(s_1 + s_2)\log n\left(\sqrt{\frac{1}{p}} + \sqrt{\frac{\log p}{n}}\right)\right) \\
&+ P\left(M(\boldsymbol{z}) \ge M(s_1 + s_2)\log n\left(\sqrt{\frac{1}{p}} + \sqrt{\frac{\log p}{n}}\right)\right) \\
\le& P\left(0 \le Q(\boldsymbol{z}) \le M(s_1 + s_2)\log n\left(\sqrt{\frac{1}{p}} + \sqrt{\frac{\log p}{n}}\right)\right) + \left(\frac{1}{n}\right)^{C_1 M} \\
&+ C_2\frac{1}{\log p} + C_3\frac{s_1}{\sqrt{p}},
\end{aligned}
$$

where $C_i$ are some positive constant. Last, from the assumption $\sup_{|x|<\boldsymbol{\delta}} f_{Q,\theta}(x) < M_2$,

$$
\begin{aligned}
\mathbb{E}\left[R(G_{\widetilde{Q}}) - R(G_Q)\right] \lesssim& (s_1 + s_2)\log n\left(\sqrt{\frac{1}{p}} + \sqrt{\frac{\log p}{n}}\right) + \frac{1}{\log p} + \frac{s_1}{\sqrt{p}} \\
\lesssim& (s_1 + s_2)\log n\left(\sqrt{\frac{1}{p}} + \sqrt{\frac{\log p}{n}}\right) + \frac{1}{\log p}.
\end{aligned}
$$

$\square$

### A.3.4 THE PROOF OF THEOREM 3.3

We first restate the convergence of the trace estimator in Gaussian setting.

**Lemma A.13.** *For multivariate normal distribution,*

$$
P\left(\left|\frac{\widetilde{tr(\boldsymbol{\Sigma}_0)}}{tr(\boldsymbol{\Sigma}_0)} - 1\right| > t\right) \le 2\exp\left\{-c\min\left\{npt^2/9, npt/3\right\}\right\}.
$$

*Proof.* Without loss of generality, assume $\boldsymbol{\mu} = \boldsymbol{0}$ . Therefore,

$$
\begin{aligned}
\widetilde{\text{tr}(\boldsymbol{\Sigma}_0)} =& \frac{\sum_{i\neq j\neq k}(\boldsymbol{X}_i - \boldsymbol{X}_j)^T(\boldsymbol{X}_k - -\boldsymbol{X}_j)}{n(n-1)(n-2)} \\
=& \frac{\sum_{i=1}^n \boldsymbol{X}_i^T\boldsymbol{X}_i}{n} - \frac{\sum_{i\neq k}\boldsymbol{X}_i^T\boldsymbol{X}_k}{(n-1)(n-2)}.
\end{aligned}
$$

For the first term, consider

$$
\begin{aligned}
\frac{\sum_{i=1}^n \boldsymbol{X}_i^T\boldsymbol{X}_i}{n\text{tr}(\boldsymbol{\Sigma}_0)} - 1 =& \frac{1}{n}\sum_{i=1}^n\left[\boldsymbol{Y}_i^T\left(\frac{\boldsymbol{\Sigma}_0}{\text{tr}(\boldsymbol{\Sigma}_0)}\right)\boldsymbol{Y}_i - \text{tr}\left(\frac{\boldsymbol{\Sigma}_0}{\text{tr}(\boldsymbol{\Sigma}_0)}\right)\right] \\
=& \sum_{i=1}^n\sum_{k=1}^p\left[\frac{\lambda_k}{n}(y_{ik}^2 - 1)\right],
\end{aligned}
$$

where $\lambda_k$ are the eigenvalue of $\frac{\boldsymbol{\Sigma}_0}{\text{tr}(\boldsymbol{\Sigma}_0)}$ and $y_{ik}$ are independent random variables from standard normal distribution.

By Assumption 3.4 and Assumption 3.5 , we have,

$$
\sqrt{\sum_{i=1}^n\sum_{k=1}^p\frac{\lambda_k^2}{n^2}} = \frac{\sqrt{\sum_{k=1}^p\lambda_k^2(\boldsymbol{\Sigma}_0)}}{\sqrt{n}\text{tr}(\boldsymbol{\Sigma}_0)} \asymp \frac{1}{\sqrt{np}},
$$

$$\sup_{k,i}\left|\frac{\lambda_k}{n}\right| \lesssim \frac{M_1}{n\mathrm{tr}(\boldsymbol{\Sigma}_0)} \lesssim \frac{1}{np}.$$

Thus by Bernstein inequality for subexponential random variables, we have for some positive constant $c$,

$$P\left(\left|\frac{\sum_{i=1}^n \boldsymbol{X}_i^T \boldsymbol{X}_i}{n\mathrm{tr}(\boldsymbol{\Sigma}_0)} - 1\right| > t\right) \le 2\exp\left\{-c\min\left\{npt^2, npt\right\}\right\}.$$

Let $\boldsymbol{Z} = \sum_i^n \boldsymbol{X}_i$. Observe that $\sum_{i \ne k} \boldsymbol{X}_i^T \boldsymbol{X}_k = \boldsymbol{Z}^T\boldsymbol{Z} - \sum_{i=1}^n \boldsymbol{X}_i^T\boldsymbol{X}_i$, with $\boldsymbol{Z} \sim N_p(\boldsymbol{0}, n\boldsymbol{\Sigma}_0)$. We first consider the concentration of $\frac{\boldsymbol{Z}^T\boldsymbol{Z}}{(n-1)(n-2)\mathrm{tr}(\boldsymbol{\Sigma}_0)}$.

$$\frac{1}{(n-1)(n-2)}\left(\frac{\boldsymbol{Z}^T\boldsymbol{Z}}{\mathrm{tr}(\boldsymbol{\Sigma}_0)} - n\right) = \frac{1}{(n-1)(n-2)}\left[\frac{\boldsymbol{Y}^T(n\boldsymbol{\Sigma}_0)\boldsymbol{Y}}{\mathrm{tr}(\boldsymbol{\Sigma}_0)} - \mathrm{tr}\left(\frac{n\boldsymbol{\Sigma}_0}{\mathrm{tr}(\boldsymbol{\Sigma}_0)}\right)\right]$$

$$= \frac{1}{(n-1)(n-2)}\left(\sum_{k=1}^p \lambda_k(y_k^2 - 1)\right),$$

where $\lambda_k$ be the eigenvalue of $\frac{n\boldsymbol{\Sigma}_0}{\mathrm{tr}(\boldsymbol{\Sigma}_0)}$, and $y_k$ are independent variable from $N(0,1)$. Similarly, we have

$$\sqrt{\sum_{k=1}^p \frac{\lambda_k^2}{(n-1)^2(n-2)^2}} \lesssim \frac{1}{n\sqrt{p}}, \quad \sup_k\left|\frac{\lambda_k}{(n-1)(n-2)}\right| \lesssim \frac{1}{np}.$$

Therefore, by Bernstein inequality, we can obtain

$$P\left(\left|\frac{1}{(n-1)(n-2)}\left(\frac{\boldsymbol{Z}^T\boldsymbol{Z}}{\mathrm{tr}(\boldsymbol{\Sigma}_0)} - n\right)\right| > t\right) \le 2\exp\left\{-c\min\left\{n^2pt^2, npt\right\}\right\}.$$

The estimation of $P\left(\left|\frac{n}{(n-1)(n-2)} - \frac{\sum_{i=1}^n \boldsymbol{X}_i^T\boldsymbol{X}_i}{(n-1)(n-2)\mathrm{tr}(\boldsymbol{\Sigma}_0)}\right| > t\right)$ follows the same process.

Combine the results above, we have

$$P\left(\left|\frac{\widetilde{\mathrm{tr}(\boldsymbol{\Sigma}_0)}}{\mathrm{tr}(\boldsymbol{\Sigma}_0)} - 1\right| > t\right)$$

$$= P\left(\left|\frac{\sum_{i=1}^n \boldsymbol{X}_i^T\boldsymbol{X}_i}{n\mathrm{tr}(\boldsymbol{\Sigma}_0)} - 1 - \frac{1}{(n-1)(n-2)}\left(\frac{\boldsymbol{Z}^T\boldsymbol{Z}}{\mathrm{tr}(\boldsymbol{\Sigma}_0)} - n + n - \sum_{i=1}^n \boldsymbol{X}_i^T\boldsymbol{X}_i\right)\right| > t\right)$$

$$\le 2\exp\left\{-c\min\left\{npt^2/9, npt/3\right\}\right\}.$$

$\square$

**Lemma A.14.** *With probability over* $1 - O(\frac{1}{p})$,

$$\|\tilde{\boldsymbol{\Sigma}}_0 - \boldsymbol{\Sigma}_0\|_{max} \lesssim \sqrt{\frac{1}{p}} + \sqrt{\frac{\log p}{n}}.$$

*Proof.* Let $t = \sqrt{\frac{\log p}{n}}$, then

$$P\left(\left|\frac{\widetilde{\mathrm{tr}(\boldsymbol{\Sigma}_0)}}{\mathrm{tr}(\boldsymbol{\Sigma}_0)} - 1\right| > \sqrt{\frac{\log p}{n}}\right) \lesssim \frac{1}{p}.$$

Follow the same process in the proof of Lemma A.5, we obtain

$$\|\tilde{\boldsymbol{\Sigma}}_0 - \boldsymbol{\Sigma}_0\|_{\max} \lesssim \sqrt{\frac{\log p}{n}} + \sqrt{\frac{1}{p}}.$$

$\square$

The subsequent proof follows essentially the same procedure as in Section A.3.3, and we present only the key steps here. For the estimators $\tilde{\mathbf{D}}, \tilde{\boldsymbol{\beta}}$,

$$\|\mathbf{D} - \tilde{\mathbf{D}}\|_F \lesssim s_1 \left( \sqrt{\frac{\log p}{n}} + \sqrt{\frac{1}{p}} \right),$$

$$\|\boldsymbol{\beta} - \tilde{\boldsymbol{\beta}}\|_2 \lesssim s_2 \left( \sqrt{\frac{\log p}{n}} + \sqrt{\frac{1}{p}} \right),$$

with a probability of $1 - O\left(\frac{1}{p}\right)$.

Next, we consider the terms in $M(\boldsymbol{z})$. For the constant term, we have

$$\left| \boldsymbol{\beta}^\top (\boldsymbol{\mu}_2 - \boldsymbol{\mu}_1) - \tilde{\boldsymbol{\beta}}^\top (\tilde{\boldsymbol{\mu}}_2 - \tilde{\boldsymbol{\mu}}_1) \right| \lesssim s_2 \left( \sqrt{\frac{\log p}{n}} + \sqrt{\frac{1}{p}} \right),$$

with a probability over $1 - O\left(\frac{1}{p}\right)$. With respect to term $\log |\tilde{\mathbf{D}}\tilde{\boldsymbol{\Sigma}}_1 + \mathbf{I}_p|$,

$$P \left( |\log |\tilde{\mathbf{D}}\tilde{\boldsymbol{\Sigma}}_1 + \mathbf{I}_p| - \log |\mathbf{D}\boldsymbol{\Sigma}_1 + \mathbf{I}_p| - \mathrm{tr}(\tilde{\mathbf{D}}\boldsymbol{\Sigma}_1 - \mathbf{D}\boldsymbol{\Sigma}_1)| \lesssim s_1 \left( \sqrt{\frac{\log p}{n}} + \sqrt{\frac{1}{p}} \right) \right)$$

$$\geq 1 - O\left(\frac{1}{p}\right).$$

Concerning quadratic term involving $\boldsymbol{z}$, follow the same process in (13), we have

$$P \left( \left|(\boldsymbol{z} - \boldsymbol{\mu}_1)^T \mathbf{D}(\boldsymbol{z} - \boldsymbol{\mu}_1) - (\boldsymbol{z} - \boldsymbol{\mu}_1)^T \tilde{\mathbf{D}}(\boldsymbol{z} - \boldsymbol{\mu}_1) - (\mathrm{tr}(\mathbf{D}\boldsymbol{\Sigma}_1 - \tilde{\mathbf{D}}\boldsymbol{\Sigma}_1)) \right| \right.$$

$$\left. \geq M s_1 \left( \sqrt{\frac{\log p}{n}} + \sqrt{\frac{1}{p}} \right) \right)$$

$$= P \left( \left| \boldsymbol{x}^T \boldsymbol{\Gamma}_1^T (\mathbf{D} - \tilde{\mathbf{D}}) \boldsymbol{\Gamma}_1 x - (\mathrm{tr}(\boldsymbol{\Gamma}_1^T (\mathbf{D} - \tilde{\mathbf{D}}) \boldsymbol{\Gamma}_1) \right| \geq M s_1 \left( \sqrt{\frac{\log p}{n}} + \sqrt{\frac{1}{p}} \right) \right)$$

$$\lesssim 2 \exp \left\{ -c \min\{M^2, M\} \right\} + \frac{1}{p}.$$

Regarding linear term involving $z$, as $(\mathbb{E}(r))^2 \geq \left(\mathbb{E}(r^{-2})\right)^{-1} = p - 2$, we can obtain

$$P \left( \left| \left(\tilde{\boldsymbol{\beta}} - \boldsymbol{\beta}\right)^T (\boldsymbol{z} - \boldsymbol{\mu}_1) \right| \geq M s_2 \left( \sqrt{\frac{\log p}{n}} + \sqrt{\frac{1}{p}} \right) \right)$$

$$\lesssim \exp\{-c_2 M^2\} + \frac{1}{p} + \frac{p}{p-2} - \frac{(E(r))^2}{p}$$

$$\lesssim \exp\{-c_2 M^2\} + \frac{1}{p}.$$

To reach the conclusion, observing that for Gaussian distribution $Q_E(\boldsymbol{z}) = Q(\boldsymbol{z})$, we consider the convergence of misclassification error.

$$
R(G_{\widetilde{Q}}) - R(G_Q)
$$

$$
= \frac{1}{2} \mathbb{E}_{\boldsymbol{z} \sim f_1} \left[ (1 - e^{\frac{Q_E(\boldsymbol{z})}{2}}) \mathbb{1} \left\{ Q(\boldsymbol{z}) > 0, Q(\boldsymbol{z}) \leq Q(\boldsymbol{z}) - \widetilde{Q}(\boldsymbol{z}) \right\} \right]
$$

$$
= \frac{1}{2} \mathbb{E}_{\boldsymbol{z} \sim N_p(\boldsymbol{\mu}_1, \boldsymbol{\Sigma}_1)} \left[ (1 - e^{-Q(\boldsymbol{z})}) \mathbb{1} \{ 0 < Q(\boldsymbol{z}) \leq M(\boldsymbol{z}) \} \right.
$$

$$
\left. \times \mathbb{1} \left\{ M(\boldsymbol{z}) < M(s_1 + s_2) \log n \left( \sqrt{\frac{\log p}{n}} + \sqrt{\frac{1}{p}} \right) \right\} \right]
$$

$$
\leq \frac{1}{2} \mathbb{E}_{\boldsymbol{z} \sim N_p(\boldsymbol{\mu}_1, \boldsymbol{\Sigma}_1)} \left[ (1 - e^{-Q(\boldsymbol{z})}) \mathbb{1} \{ 0 < Q(\boldsymbol{z}) \leq M(\boldsymbol{z}) \} \right.
$$

$$
\left. \times \mathbb{1} \left\{ M(\boldsymbol{z}) < M \log n (s_1 + s_2) \left( \sqrt{\frac{\log p}{n}} + \sqrt{\frac{1}{p}} \right) \right\} \right]
$$

$$
+ P_{\boldsymbol{z} \sim N_p(\boldsymbol{\mu}_1, \boldsymbol{\Sigma}_1)} \left( M(\boldsymbol{z}) \geq M \log n (s_1 + s_2) \left( \sqrt{\frac{\log p}{n}} + \sqrt{\frac{1}{p}} \right) \right).
$$

Combine the results above, we can reach the conclusion that

$$
\mathbb{E} \left[ R(G_{\widetilde{Q}}) - R(G_Q) \right] \lesssim \left[ \log n (s_1 + s_2) \left( \sqrt{\frac{\log p}{n}} + \sqrt{\frac{1}{p}} \right) \right]^2 + \frac{1}{p}
$$

$$
\asymp (s_1 + s_2)^2 \log^2 n \left( \sqrt{\frac{\log p}{n}} + \sqrt{\frac{1}{p}} \right)^2.
$$

$\square$

