# OpenReview forum: "A Spatial-Sign based Direct Approach for High Dimensional Sparse Quadratic Discriminant Analysis"
_ICLR.cc/2026/Conference — ICLR 2026 Conference Withdrawn Submission_

### Official Review · Reviewer_kwLw · 2025-10-26

**Soundness:** 2
**Presentation:** 3
**Contribution:** 2
**Rating:** 4
**Confidence:** 4

**Summary:**

The paper studies high-dimensional sparse quadratic discriminant analysis (QDA) and proposes a new classification method, SSQDA, using constrained convex optimization based on the sample spatial median and spatial sign covariance matrix under elliptically symmetric distribution assumptions. The classifier achieves an optimal convergence rate, and simulations and real data applications show it is robust, especially for heavy-tailed distributions, offering practical benefits for high-dimensional data analysis.

**Strengths:**

1.The authors introduce a novel approach to high-dimensional sparse QDA by leveraging constrained convex optimization based on the sample spatial median and spatial sign covariance matrix.

2.The proposed model can achieve the optimal convergence rate over a broad class of parameter spaces, up to a logarithmic factor.

3. Simulation studies and real-data applications are used to show that SSQDA is robust and efficient.

**Weaknesses:**

1.SSQDA depends on the assumption of an elliptically symmetric distribution.
2. The authors derive the model based on two-class problems. It is important to explore the model corresponding to multi-class problems.
3. The paper does not explicitly mention the computational complexity of SSQDA. Constrained convex optimization problems can sometimes be computationally intensive, especially in high-dimensional settings.
4. The effectiveness of SSQDA may depend on careful tuning of its parameters, such as the choice of the constraint set in the convex optimization problem.
5.The authors explore robust means and covariances in discriminant analysis. Robust discriminant criteria are used to deal with the heavy-tailed distributions.  What is the difference between them for heavy-tailed distributions?
6.The experiments on real data sets are limited.

**Questions:**

How to perform  SSQDA  on large-scale data sets?

---

### Official Review · Reviewer_8yA5 · 2025-10-28

**Soundness:** 3
**Presentation:** 3
**Contribution:** 3
**Rating:** 4
**Confidence:** 3

**Summary:**

This paper investigates the problem of sparse quadratic discriminant analysis (SQDA) in high-dimensional settings. Traditional QDA methods fail when dealing with high-dimensional data, especially when the data do not satisfy the normality assumption. Existing high-performance methods are mainly designed for Gaussian distributions and may perform poorly in the presence of heavy-tailed distributions or outliers. To address this issue, the authors propose a new classification method called SSQDA (Spatial Sign-based Sparse Quadratic Discriminant Analysis).

**Strengths:**

* The core innovation of the paper lies in the creative integration of the concepts of "spatial sign" and "spatial median" from robust statistics into the direct estimation framework of high-dimensional sparse QDA. This idea is quite innovative: instead of simply replacing estimators, it provides a robust solution based on convex optimization for this classic problem, effectively breaking through the reliance of traditional methods on the normality assumption.

* he authors have established a comprehensive theoretical framework for the proposed SSQDA method. Notably, the paper not only presents the convergence rates of key parameter estimators but also derives the convergence bounds of misclassification error under the more general elliptically symmetric distributions. Theoretical analysis shows that the method achieves a rate comparable to mainstream approaches under normal distributions and exhibits significant robust advantages under heavy-tailed distributions, providing a solid theoretical guarantee for its effectiveness.

* Through systematic simulation experiments and real data validation, the paper comprehensively evaluates the method’s performance. The experimental design covers various dimensions, covariance structures, and important non-normal distribution scenarios. Consistent results demonstrate that SSQDA performs excellently when assumptions hold and shows even more prominent advantages when assumptions are violated, making the conclusions highly persuasive.

**Weaknesses:**

* Rigorousness of Innovation Description: The Boundary with Existing Works Needs to Be ClearerWhen emphasizing its originality, the paper mainly compares with methods based on the normality assumption (e.g., Cai & Zhang 2021). However, the work by Lu & Feng (2025) cited in the paper ("Robust sparse precision matrix estimation") seems to have already applied spatial sign methods to graphical models and sparse LDA. The authors claim that SSQDA is the "first systematic extension," but they have not fully clarified the essential differences and connections between their work and that of Lu & Feng (2025) in terms of sparse LDA.

* Comprehensiveness of Experimental Comparisons: Lack of Key Robust Baseline MethodsThe experimental section compares SSQDA with SDAR, SLDA, LDA, and QDA, most of which are sensitive to outliers. However, to fairly demonstrate the "robustness" advantage, the experimental design should include other types of robust classifiers as baselines.

* Details on Practical Application and Parameter SelectionThe paper mentions that the regularization parameter is selected via cross-validation, but key details are lacking. In high-dimensional settings, cross-validation incurs high computational costs and its results may be sensitive to data splitting. Additionally, for real-world data applications, readers cannot assess the method’s practical usability.

**Questions:**

* The paper repeatedly cites and relies on the core lemmas (Lemma A.2, A.3) of Lu & Feng (2025) (arXiv:2503.03575). We kindly ask the authors to clarify more explicitly: What are the essential methodological differences between the robust sparse precision matrix estimator proposed in Lu & Feng (2025) and the core method in this paper? Should the contribution of this paper be more accurately positioned as innovatively applying and systematically extending the estimation framework of Lu & Feng (2025) to the high-dimensional QDA problem, along with establishing a complete misclassification theory for it—rather than presenting a completely new estimation method developed from scratch?

* Did the authors perform any data preprocessing (e.g., mean centering, scaling)? SSQDA relies on the spatial sign U(X) = X/||X||₂, which means it is insensitive to the global magnitude of each sample but may be sensitive to the relative intensity distribution between pixels. Additionally, image data typically exhibit spatial local correlation, yet this method does not appear to leverage such structure.

* Could the authors provide some qualitative or quantitative comparison of the practical runtime between SSQDA and SDAR? Furthermore, is the variation range of parameters c₁ and c₂ large across multiple experiments? And is it stable?

---

### Official Review · Reviewer_Nbe9 · 2025-10-31

**Soundness:** 3
**Presentation:** 4
**Contribution:** 2
**Rating:** 4
**Confidence:** 3

**Summary:**

This paper proposes a new classifier called SSQDA for high-dimensional data where the number of features is larger than the sample size. The method is designed to handle heavy-tailed distributions and outliers that often cause traditional classifiers to fail. SSQDA uses robust spatial sign-based estimators instead of conventional means and covariances, making it resilient to non-normal data. It employs sparse optimization to directly estimate the key parameters needed for classification. Theoretical analysis shows the method achieves a near-optimal convergence rate for its estimation error and misclassification risk. Some simulations demonstrate that SSQDA outperforms existing methods under heavy-tailed and high-dimensional settings. The approach is also validated on one example of a real-world image classification task for detecting concrete cracks, where it achieves slightly superior performance.  This work aims at providing a robust and theoretically grounded framework for high-dimensional classification, especially when data deviates from Gaussian assumptions.

**Strengths:**

On the theoretical front, the paper presents work appears  robust and substantial. The core contribution lies in the mathematical development, which seems to be the strongest and most innovative element of the manuscript, although it heavily builds on (previously obtained) recent results.

 The body of the text, particularly the explanations surrounding the theorems, is relatively easy to read and well-explained, which aids in understanding the technical developments.

 The proof technique employed could bring genuine value to the scientific community by enriching the methodological toolbox for this type of complex, high-dimensional statistical analysis. The claimed generality of the results, which appear to hold without loss and apply without any counterexamples, even toy ones, is also a notable point, albeit a surprising one.

**Weaknesses:**

- an important part of the proof seems to be recently published in a recent preprint (Lu and Feng).

- On the methodological and experimental side, the algorithmic novelty is perceived as limited. The main change involves using the Spatial Sign Covariance Matrix (SSCM) as a plug-in replacement for classical estimators in the LDA/QDA frameworks, an idea that has been seen before in the literature, probably even in papers on discriminant analysis.

- Some recent literature on robust estimation for LDA/QDA might be missing:
For instance (and see several papers in the same vein):
Houdouin, P., Wang, A., Jonckheere, M., & Pascal, F. (2022, May). Robust classification with flexible discriminant analysis in heterogeneous data. In ICASSP 2022-2022 IEEE International Conference on Acoustics, Speech and Signal Processing (ICASSP) (pp. 5717-5721). IEEE.


The experimental section seems light for a top-tier conference, with only one real dataset used to validate the method. This choice makes it difficult to observe clear and robust trends in the comparisons with the numerous LDA and QDA variants. Finally, the experimental setup does not seem designed to demonstrate impressive performance that could compensate for the perceived lack of algorithmic innovation.

**Questions:**

- It is particularly surprising to see such general results claimed with the SSCM, given that it is a biased estimator.  Can you comment a bit more on this?

- the theoretical results are valid for any number of classes?

- the results on real data are not especially impressive in terms of performance gain? did you try on other datasets?

- why the same scale r.v. for the two classes?

- the references are not numbered, some quoted references are missing?

---

### Note · Authors · 2025-11-17

I have read and agree with the venue's withdrawal policy on behalf of myself and my co-authors.